# Added Value on a Day in the Pandemic in Tourist Attractions in the Polish–Czech Borderland as a Green Economy Initiative

Małgorzata Pol *, Małgorzata Rutkowska and Jerzy Tutaj *

Faculty of Management, Wroclaw University of Science and Technology, 50-370 Wroclaw, Poland
* Correspondence: malgorzata.pol@pwr.edu.pl (M.P.); jerzy.tutaj@pwr.edu.pl (J.T.)

**Abstract:** This study documents customer behavior in the travel services market before and during the COVID-19 pandemic (C-19). We offer theory-based and research-based insights that demonstrate customer value propositions during a pandemic and help predict future behavior for green tourism development. This article aim is to identify the relationship between the situation in tourism during the pandemic, customer behavior, and the added value that historical sites in Poland and the Czech Republic introduced or should introduce in the green economy. The topic is important because the situation during the pandemic showed the lack of a quick response, which is only possible if you have prepared scenarios for the crisis. This study discusses the marketing factors of creating value and analyzes the value strategy for individual clients. The authors of this study analyzed various stages in the customer life cycles in the company and the level of their profitability, taking into account the principles of the green economy (sustainable development) in the example of cultural facilities, i.e., Książ Castle in Poland and the Kuks Complex in the Czech Republic. Literature studies were used in this study, followed by the questionnaire method. The test results are presented in tabular form and supplemented with graphical forms.

**Keywords:** added value; COVID C-19; green economy; tourist attractions

## 1. Introduction

This article's aim is to identify the relationship between the situation in tourism during the pandemic, customer behavior, and the added value that historical sites in Poland and the Czech Republic introduced or should introduce in the green economy. The topic is important because the situation during the pandemic showed the lack of a quick response, which is only possible if you have prepared scenarios for the crisis. Modern times are often defined as a time of permanent crisis. Therefore, it is crucial for those running companies today, including heritage managers, to constantly observe how to improve and create new added value to retain and/or acquire customers. On the other hand, the research gap is to identify those added values for clients of historical sites that should be introduced regardless of the pandemic period due to permanent changes in client behavior and expectations. The added values that the authors observed, as activities introduced or desired at both historic sites not only during the pandemic, are as follows:

1. Increasing the safety of personal hygiene and interpersonal contacts through the introduction of tools such as the development of a particular instruction manual;
2. Paying attention to green management, which results from savings, as well as customer expectations;
3. Using tools from the area of digitization both when selling services as well as enriching the product, distribution, and promotion of services in the two historic sites.

In the coronavirus era, the consumer's attitude has changed because his activities related to market activity were influenced by a different way of thinking and assessing the situation. The current lifestyle, focusing on the problems of frequency of shopping, participation in social circles, entertainment, recreation, hobbies, or the very attitude to

work, had to change, regardless of the will of a given person. This surprising situation changed customers' personalities, which, as it was found, under stable conditions, changes over the years, and today's time has accelerated and changed us, the environment, and even our life goals, plans, and priorities. Marketing changes behaviors, habits, views of individuals and social groups, and the value system under the influence of specific activities, called marketing activities. Practical marketing activities are now considered to have a real, measurable impact on the financial result, profitability, or value of the enterprise, especially by increasing the total customer value [1]. New marketing trends are reflected in social, cultural, and ecological aspects resulting from buyer behavior changes, which new organizations should remember. As shown by market laws, enterprises cannot only focus on business goals but must also consider the principles of the green economy. A green economy "means an idealistic concept, but painstakingly implemented in practice, which aims to ensure an increase in welfare and quality of life as well as social equality while stopping the depletion of natural resources and limiting ecological threats" [2].

Moreover, this slogan is eagerly used by non-governmental organizations. Therefore, the company's goals should be integrated with social and environmental objectives in a way that contributes to solving social problems, which, in turn, will bring marketing profits, translating into final financial results and building a competitive advantage. So far, strategies, business models, advertising campaigns, and other promotional campaigns have helped these efforts. Efforts, as already mentioned, now need to be verified because the coronavirus has caused considerable changes in people's existing and proven mechanisms and living standards worldwide and does not intend to stop.

Much of the current marketing efforts forced by the COVID-19 pandemic will be permanent, resulting in long-term changes in companies' marketing strategies. Understanding the essence of the customer's role requires a clear distinction between value for the customer and customer value. There are many attempts in the literature to define the concepts mentioned above, which prove the existence of a significant variety of views, positions, and interpretations. Ph. Kotler proposed the following definition of value: "The value delivered to the customer is the difference between the total value of the product for the customer and the cost that the customer must incur in obtaining it. The total value for the customer is the sum of the benefits that he expects from a given product or service" [3]. According to the author of the paper, value for the customer is a designed process consisting, on the one hand, of satisfying the specific needs of buyers by delivering satisfaction more efficiently and effectively than competitors could and, on the other hand, ensuring the achievement of added value for the individual. P. Doyle, speaking in the discussion on the marketing factors of creating value, claims that there are three principles of creating usability for the customer: first, the buyer must be convinced that when choosing a given product or service, he chooses the company that feels offers him the highest value; second, it should be realized that the customer is not mainly motivated by the product or service, but by the possibility of satisfying his needs through them; and third, there is a need to move away from focusing on short-term transactional thinking towards long-term relationship marketing. As for the second of the concepts mentioned above—customer value—it is a bundle of value (added value) provided by buyers to the enterprise, e.g., revenues (through customer satisfaction), information and signals, trust, loyalty and friendship, and other values. Therefore, it is correct to say that companies are "customer-driven". In practice, companies that want to maximize customer value for the enterprise must take the following actions: STP (Segmentation Targeting Positioning) process and building individualized value strategies for individual customers, and taking into account different stages in the company's customer life cycles and their profitability level.

The company's success, measured by the increase in market value, is maximizing profit by providing value for customers. Without a deal for buyers, there is no value for the shareholder, and value for shareholders correlates with value for customers. Social isolation is currently influencing the differentiation of the value system, which will then translate into the differentiation of our decision-making methods and behavior in the

market. In effect, it will lead to the emergence of new business models, forcing some industries and enterprises to adapt faster to different customer expectations. It can be expected that the process of changes will take place quickly, as companies in 2020 have undergone and are currently going through a severe crisis and will be anxious to reactivate their activities as soon as possible. This situation primarily applies to the tourism market. We observe changes in offers, customer service, levels and types of quality of products and services offered, and the emergence of new business models. Observing and considering scenarios for the development of the situation in the leisure industry, i.e., those industries that relate to how people spend their free time and at the same time spend the money saved due to work. These include culture, entertainment, recreation, sport, and, above all, tourism. In these spheres, the situation is extremely worrying, but some are already making some tourism industry preparations. This article discusses these changes and enterprises' position in the historical tourism field. The authors of this article noticed the enormous changes in the tourism sector, where during the coronavirus pandemic, the managers of tourist facilities are obliged to properly correlate their offer to new consumer requirements and restrictions imposed by the governing bodies in a given country. The top-down regulation of activities for tourist facilities meant that the centers' managers began to look for new solutions and possibilities for further provision of services. The tourism industry is willing to guarantee adequate safety and high sanitary conditions for visitors and maintain the financial liquidity of the conducted activity; the tourist offer should be modified. Figure 1 below graphically shows the relationships between tourist facilities, clients, and governmental restrictions. Bidders wishing to reach the broadest possible group of consumers should constantly exchange information so that the offer can be almost tailor-made. The pandemic has helped to shake up this relationship. Nowadays, the third entity, the government, cannot be overlooked, which in crises, can change the conditions of the environment for both customers and bidders—changing the current expectations of the actors. The flow of information should be continuous so that the offer is as tailor-made as possible to meet customer expectations. The pandemic has shown a new face to the relationship between demand and supply. Government restrictions on the public, as well as on the service provider, have shaken this exchange. A new perspective on creating a tourist offer must be needed to consider external independent factors.

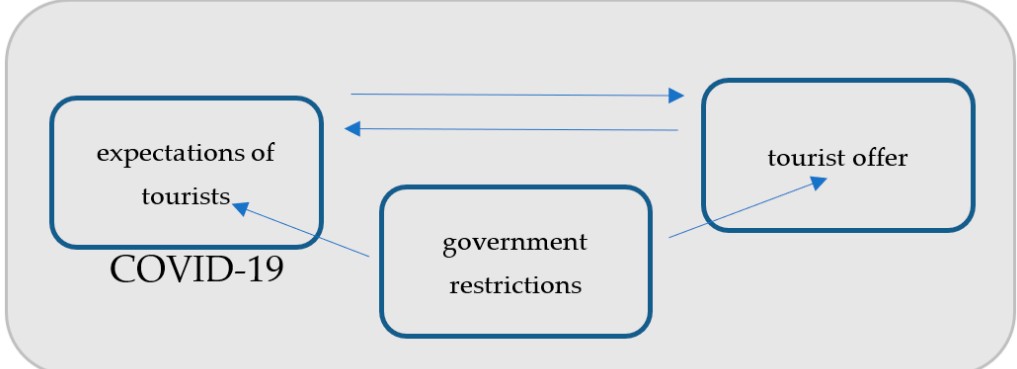

**Figure 1.** Tourism service marketing relationships during the COVID-19 pandemic. Source: own study.

In the new environment that the whole world has found, the environment of the coronavirus pandemic, the category of customer value is understood broadly in the theory of marketing, but there is no clear definition of it. Generally, it relates to the consumer's life in a given environment and the needs they satisfy. According to the theoretical assumptions of modern marketing concepts, for example, relationship marketing [4], value marketing [5], and marketing 3.0 and 4.0 [6], the product (or, more broadly, the offer) has a specific value for each consumer. In economics, this category is understood in a narrower sense, related to utility (the so-called use value), i.e., the ability of a product to meet specific consumer

needs. The buyer judges it based on the perceived benefits of purchasing and using the product [7]. When analyzing the definitions of value for the customer, it can be indicated that it includes a set of benefits obtained by the customer (e.g., functional, emotional, cognitive, and/or social). Those benefits are related to objective and subjective attributes of a given product (offer) compared to the costs (financial and non-financial, e.g., time, effort/energy, mental costs) incurred for its purchase and use. This value is a subjective reflection of the level of a given product's ability (offer) to meet a specific need or, more often, a bundle of needs of a given consumer throughout his life cycle at the customer. When comparing the definitions of value for the customer, one can also notice differences in understanding this category. They mainly concern the type of value. The research, and as a result, the formulation of many definitions of value for the customer, was inspired by the concept of the added value chain developed by M. Porter [8]. The category of customer value is recognized as one of the key ones in marketing. Its fundamental importance results from the value of the product (the offer), or rather its subjective perception by the consumer. This is a crucial element in the process of making purchasing decisions (each buyer may perceive the value of a given offer differently and, as a result, make different decisions). This value determines the size and structure of the demand and, as a result, the company's competitive position. Value creation for the customer by the company takes place within a specific system. It includes defining value for the customer, shaping a value proposition tailored to the customer's needs, communicating and delivering the value proposition to the customer, and overseeing the use of the value received by the customer. This system also covers all the buyer's experiences until the value is obtained while using the product. The company should adequately manage such a system so that the client's value is, in its opinion, the highest and exceeds the value offered by competitors. It is worth emphasizing that the company (producer and seller) can shape and/or create consumers' needs and expectations of the product it offers with the help of specific strategies and marketing tools within them. This way, it influences customers' assessment of the final value obtained by purchasing and using a given product. Companies should act according to the "4A" concept when creating value for the client. It takes into account the general values sought by buyers:

- Acceptability, which refers to the degree to which the offer meets or exceeds the consumer's expectations;
- Affordability, which is related to the degree to which target segment consumers are able/willing to pay the proposed price for the product. Two dimensions can be indicated here, i.e., economic availability (buyer's ability to pay) and psychological availability (buyer's willingness to pay).

Expected value is a surplus of benefits subjectively perceived and expected by the consumer concerning the expected costs related to the purchase and use of the product offered on the market. On the other hand, the value obtained should be understood as a surplus of the subjectively perceived benefits received by the customer over the perceived costs incurred by him to purchase and use the product [9].

The e-marketing tools in the creation process include the following:

- Accessibility, which covers the degree to which consumers are willing to buy and use a product; there are two dimensions here, i.e., product availability and convenience when purchasing and using;
- Awareness, which refers to the degree to which the bidder informs consumers about the product's characteristics, induces them to try it, and encourages them to buy again; two dimensions can be distinguished here, i.e., brand awareness and product knowledge.

The basis of market exchange is value. The buyer selects the offer by assessing the manner of satisfying the needs and estimating the satisfaction achieved after its purchase. A product that better meets needs and gives greater satisfaction will be assigned more significant value. The evaluation of value will rarely be rational, mainly due to the consumer's lack of complete information and reluctance to collect comprehensive data about

the offer and its availability, and also due to the subjective perception of individuals. Nevertheless, to reduce the mental discomfort that may arise after the purchase, the buyer will choose those goods and services that, in his opinion, are of the highest value. Value for the customer is the basis for shaping the relationship between the company and the buyer. B. Dobiegała-Korona notices that the main task of a marketing strategy is to build value for customers, which they will find satisfactory, and thus they will stay with the company for a long time while making the relationship profitable. Satisfaction is a derivative of value for the customer. Contentment is the basis of regaining loyalty and sometimes even complete loyalty.

## 2. Material and Methods

Tourism is among the economic activities that have been particularly affected by the pandemic caused by the COVID-19 pandemic [10]. According to UNCTAD [11], the collapse of international tourism due to the pandemic will result in losses of more than $4 trillion in global GDP in 2020 and 2021. Tourism's extreme vulnerability to crisis events is due to the global interdependence among its players (businesses and countries) and the huge role of traveler safety [12,13]. The scale of change and the impact of the pandemic make it possible to classify the event as a metaphorical black swan. This term is attributed to events of great significance and serious consequences that dramatically change the political and economic environment [14]. Black swans can also present opportunities for entities that correctly read market signals and adapt to new business conditions. Finally, it is also the kind of event that has triggered the need for a fundamental change in the ways of conducting economic science in both ontological, epistemological, and methodological terms [15]. However, the COVID-19 pandemic caused an unprecedented decline in tourism and enormous losses for the entities that make up the tourism economy [16]. The consequences of the pandemic for the tourism industry can be considered in terms of primary quantitative and secondary qualitative changes. As a direct consequence of the spread of the coronavirus, tourism activity was reduced due to the need for social distance and isolation. As a result of the restrictions imposed by the governments of most countries, tourist traffic practically froze in 2019 and fluctuated in the following months in response to the resilience of startups to the crisis caused by the COVID-19 pandemic, to successive periods of relative normality and consecutive waves of intensification of the pandemic and related restrictions. The effect of the reduction in tourism was secondary to qualitative changes associated with the adaptation of tourism economy players to changing buyer expectations and imposed formal limits. Fear of distant travel brought a renaissance to domestic tourism and so-called mini-travel. The need for social distance has led to a change in expectations of accommodations and an increase in interest in facilities such as motor homes, apartments, and RVs. The popularity of individual means of transportation has also increased, making it possible to reduce social contact. An important qualitative change brought about by the pandemic was the acceleration of labor substitution processes through technology. In gastronomy, ordering, payment, and delivery processes were carried out using apps and websites, and the importance of cashless payments increased. In the context of the indicated changes, the problem of the resilience of tourism enterprises (resilience), understood as the ability of an organization to maintain and adapt its basic structure and functions in the face of disruptions, becomes important [17]. The concept of resilience helps to understand how tourism organizations can effectively respond and positively adapt to a changing environment. Pforr and Hosie [18] believe that crisis management should broadly seek to anticipate possible crisis events that may occur (pre-crisis phase) to reduce or mitigate the effects of the crisis (crisis phase), and to quickly, effectively identify and repair the damage caused by the crisis (post-crisis phase). In the current environment of deep uncertainty, entrepreneurs need to recognize both existing risks and emerging opportunities to understand changes in society's and the market's needs [19]. Jiang, Ritchie, and Verreynne [13] believe that one of the key determinants of tourism businesses' resilience to crisis events is their dynamic capabilities. They make it possible to create new or modify

existing resources and organizational routines, recognize and exploit opportunities inherent in the environment, and maintain or improve the ability to compete in a dynamic environment. Dynamic capabilities can be defined as "the ability to integrate, build and reconfigure internal and external competencies to recognize a rapidly changing environment" [20] or "specific strategic and organizational processes such as product development, alliances and strategic decision-making that create value for companies in dynamic markets" [21]. Dynamic capabilities help organizations adapt to a changing environment by using slack resources and transforming operational routines to achieve resilience to crises [22]. Jiang, Ritchie, and Verreynne [13] believe that the key competencies of enterprises most often include the ability to continuously modify operational routines, proactive learning capabilities, organizational leadership styles, networking, and collaboration. Research in Germany shows that one of the key resources used by startups to cope with the COVID-19 crisis is relational resources, which would include favorable partners, mutual support in the startup community, and access to social capital [23]. Other key resources identified by startup founders include financial resources, which come from internal resources or from applying for government support or venture capital funds [24].

Under current marketing concepts, companies are trying to tailor their activities to select target consumer groups. These concepts assume that a company's activities aim to determine the needs of consumers from the selected target groups and then satisfy them as best as possible. Consumers discover their needs during the first stage of the decision-making process, which is problem recognition. The following sets of the purchasing decision-making process are related to the consumer's decision making related to satisfying their needs. During the process, each consumer exhibits certain behaviors. These behaviors are called consumer behavior in the literature. Consumers, which are people, constantly have to make decisions regarding purchasing goods. The decisions they make are significantly influenced by their behavior. In the literature, one can find many approaches to defining consumer behavior and its scope (Table 1).

**Table 1.** Definitions of the term "Consumer Behaviour".

| No. | Author | Definition |
|---|---|---|
| 1 | Kotler P., Keller K.L. | "Consumer behavior is an area of study that includes how consumers, groups, and organizations select and purchase goods, services, ideas, and experiences to satisfy their needs and wants, and how they use and dispose of them." [25] |
| 2 | Falkowski A., Tyszka T. | "Consumer behavior includes everything that occurs during and follows the consumer's purchase of goods and services" [26] |
| 3 | Jachnis A. | "Consumer behavior is the study of buying behavior and exchange processes involved in buying, consuming, and disposing of products, services, and ideas." [27] |
| 4 | Solomon M., Bamossy G., Askegaard S., Hogg M.K. | "Consumer behavior includes the study of the processes of individuals or groups who choose, buy, use, and dispose of, service products and/or services, as well as the study of how individuals or groups satisfy their needs" [28] |
| 5 | Stasiuk K., Maison D. | "A field of science concerned with understanding humans in a market environment" [29] |

Source: own study.

An important task is to analyze and understand consumer behavior, which is necessary to know the factors that shape consumer behavior and decisions. Among them, the following three groups of elements can be distinguished: cultural, social, and personal [25]. Consumer behavior is defined as "the totality of the consumer's actions and perceptions that make up the preparation of the decision to choose a product, make that choice, and consume" [30]. The decision-making process in tourism involves many decisions regarding the trip's destination and the product's components. Decisions are influenced by a wide variety of often interdependent factors. As purchasing funds increase, the scope of consumer decisions expands, and aspects of a psychological and social nature influence decisions. These factors condition the undertaking of tourist activities and, through their influence

on the buying and selling processes in the tourist market, affect the formation of tourist demand. The impact of non-economic determinants of consumer behavior justifies the need for an interdisciplinary approach to research and the inclusion of an analysis of tourist behavior from a sociological and psychological point of view. The behavior (conduct) of the consumer in the tourism market is the totality of his reactions to various types of stimuli, both internal (motives, needs) and external (environment), appearing in connection with the satisfaction of tourism needs using consumption (material goods and services) [31]. They are also called consumption (consumer) behaviors, and Zabinskaya [32], emphasizing the specificity of tourist needs, calls them tourist behaviors. A distinction is made between two types of consumer behavior: market behavior (the totality of the consumer's activities comprising the preparation of the decision to choose a product, the making of this choice, and the purchase) and consumer behavior in the consumption phase (the activities of using and utilizing the means of consumption directly in the acts of satisfying needs). These can be supplemented by post-consumption behavior—conduct towards the residues of consumption (used goods, garbage, waste) and conduct motivated by the effects of consumption (satisfaction, dissatisfaction, memories).

The main objective of the conducted research was to identify the added value that was generated or should be implemented by companies managing historical sites in both Poland and the Czech Republic during the pandemic, based on internal documents of both organizations, analyses of the level of tourist traffic, interviews with managers and surveys among tourists, which both companies carry out on an ongoing basis. The authors also conducted their research through face-to-face interviews among tourists at both facilities. The poll took 132 respondents between June and August 2020 at both facilities.

The research problems focused on the following:

1. Determining the quantitative relationship between pandemic-related insults at different times of the pandemic and the level of ticket sales;
2. Identifying the response of businesses to changes resulting from the pandemic in general and concerning green economy solutions;
3. Identifying the level of customer satisfaction with the changes made;
4. Identifying the added value accepted and postulated by customers of both monuments;
5. Identifying differences in the practice of the two monuments in two different countries during the pandemic;
6. Identifying the added value for the customer in both monuments as a recommendation for other such facilities in Poland and the Czech Republic concerning the green economy.

In connection with the development of the problem areas, the authors identified the following research questions:

1. How did the level of visitors change with the pandemic at the two sites?
2. How have changes in the level of ticket sales affected the overall operation of the two facilities?
3. How was both facilities' adaptation and response process during the pandemic?
4. What measures have been taken at both facilities to maintain financial stability in connection with the pandemic? What actions have been taken by both facilities to change competitive instruments (product, price, distribution channels, promotional activities)?
5. How did both facilities react to their customers during the pandemic? With particular attention to solutions relating to the green economy, what resources did both facilities tap into?
6. What lasting changes in buying behavior and expectations did the pandemic cause among customers of historic sites?
7. What added value did both venues introduce during the pandemic, and what added value did their customers expect?
8. What recommendations can be defined for similar facilities in both countries at the level of such crises emerging? Additionally, which recommendations can be adopted permanently as a continuous improvement in their services?

During this study, the authors used the following research methods: simple statistical methods, comparative analysis, analysis of foundational documents in both entities, interviews with the management of both facilities, and discussion and analysis of surveys conducted.

## 3. Discussion

Most of the available research results on consumer behavior resulting from the onset of a pandemic concern the first few months after the outbreak. They cover selected aspects of purchasing behavior in various product categories. The results of studies conducted in the early period of the pandemic indicated that consumers changed their behavior and sharply increased their purchases of certain products [33,34]. As M. Loxton et al. [35] point out, consumers in the first months of the epidemic behaved similarly to previous crises, such as during the 2007–2009 financial crisis; they bought products in a panic, reduced spending on elective goods, and made their decisions based on media reports.

The COVID-19 pandemic has significantly affected tourism. Many places that enjoyed enduring popularity for years saw a significant drop in visitors in 2020. The introduction of restrictions in the form of a ban on movement has contributed to a substantial decline in tourist arrivals in Poland and the Czech Republic. The literature describes the changes that COVID-19 has caused in the tourism economy [36], formulates multivariate forecasts of the development of the situation, draws scenarios for recovery from the crisis, and proposes corrective measures. A Special Issue of Tourism Geographies [37] was published, containing as many as 25 articles on COVID-19-related topics, for example, outlining a vision for post-emergency tourism [38], proposing necessary economic measures to save the tourism economy [39], and pointing to the required transformation [40]. There are theses about the need to socialize and green tourism after COVID-19 [41,42], with a strong emphasis on sustainability and responsible tourism [43]. It has been proven that during recovery from an epidemic crisis, traveling within one's own country and staying at agritourism farms and facilities to ensure sanitary safety will become more popular [44]. In the era of pandemics, business models in tourism, in the area of sights, referred to the opportunities offered by the virtual world. We are seeing more and more discussions related to changes in the progressive digitization of sales processes [45,46]. Researchers recognize a gap in the area of understanding the impact of Artificial Intelligence (AI), including Machine Learning (ML) and Digital Marketing (DM), on modern sales [45,47]. The literature contains numerous studies on the distinction between marketing and sales [48].

In contrast, research on how consumer behavior changes affect the sales function's remodeling relative to the marketing function is an episodic phenomenon [49]. Currently, the biggest challenge facing businesses and management trends is the COVID-19 pandemic, which affects all business areas, including the sales function [50]. A pandemic outbreak is part of a broader problem of various phenomena referred to as exogenous shocks, which tend to have a jumping effect on changes in economic and social processes [51]. They are part of the peculiarities of standard economic shocks: demand, supply, price, and financial shocks have affected almost all industries, forcing companies to reformulate their implemented strategies, including sales strategies [48,52–54]. Knowledge in this area is drawn mainly from commercial publications, consulting firms' reports, and practitioners' studies [55–57]. Overall, during the COVID-19 pandemic, a theory can be advanced that the tourism system is more vulnerable than other systems [58]. The pandemic caused unprecedented effects: it affected tourists' lifestyles, behaviors, and travel patterns [59]. In addition, an overall increase in mental disorders caused by isolation, such as increased anxiety, impacted the frequency and form of tourism participation [60,61]. The pandemic also resulted in other adverse consequences, such as creating a negative image of travel [62]. Quite a bit of change could also be seen in the destination's vision, which, as is well known, can change constantly. The COVID-19 pandemic significantly impacted its further formation [61]. Overall, it affected global supply and demand [63]. Hence, it is understandable that there has been a marked increase in interest in the issue of the tourism crisis and crisis

management [40,64], among others, in the context of pandemic mitigation and post-crisis recovery of the industry [65].

In the Polish literature, one can find studies of a pilot and non-representative nature [66,67]. However, they allow us to see symptoms of changes in the structure of spending (reduction in clothing and footwear purchases), increased online shopping activity, and the use of instant messaging in the purchasing process. A change in consumer attitudes has been identified: the growing importance of rest, relaxation, and a healthy lifestyle. In the pages of the journal Organization Review, the results of the study of consumer behavior in light of COVID-19 have not yet been published, but the issues of sustainability of behavioral changes under the influence of modification of marketing activities of enterprises and habits of buyers have been considered [68]. The author of the cited article put forward the thesis that the changes observed during the pandemic will be permanent.

When considering the tourist attractions of the Polish–Czech border region as a green economy initiative, it is necessary to point out what the green economy is. The concept has emerged in debates about sustainable development, especially in light of the Rio +20 idea [69]. For the aims of this article, it has been assumed that the green economy generally refers to an economy in which the quality of human life and the state of the environment are broadly considered when making production and consumption decisions. Hence, the inclusion of tourist attractions of the Polish–Czech borderland in this concept/initiative should be considered a novelty [69].

## 4. Analysis of the Market Situation of Both Studied Objects in the Period Preceding the Pandemic and during the Pandemic

During the pandemic, many freedoms were limited to society, and at the same time, the duration of the lockdown initiated the creation of a new face for the client. As the government introduced many obligations, including those related to the form of communication, some companies started the era of business meetings via the Internet. At that time, the Internet's power and broadly understood digitization became almost the fundamental tool in performing official duties. Communication via platforms such as ZOOM or Teams is long enough to affect current consumer habits. This thesis seems to be confirmed by a survey conducted among customers visiting the analyzed cultural sites. The analyses were carried out in the first stage to verify the impact of the ongoing coronavirus pandemic on the attractiveness of various age groups' tourist facilities.

Table 2 below presents the number of tourists who visited the analyzed cultural site, Książ Castle, during the summertime, i.e., in July, August, and September, in the last three years.

**Table 2.** The number of tourists visiting cultural sites in the summer of 2018–2020.

| The Number of Visitors to Książ | July | August | September |
|---|---|---|---|
| 2018 | 68,000 | 91,000 | 49,000 |
| 2019 | 72,000 | 93,000 | 51,000 |
| 2020 | 58,000 | 71,000 | 37,000 |

Source: own study based on data obtained from the Książ Castle Company in December 2020.

Table 1 above shows that in the summer period, the most significant number of visitors to the Castle year after year falls in August. The year 2020, which is a difficult time for enterprises in the field of tourism, saw a downward trend in tourist traffic in the summer compared to August in previous years. Despite declining activity among tourists visiting the Castle, the number of people who purchased tickets during the SARS-CoV-19 pandemic in August was 71,000. This value is similar to the number of tourists visiting the analyzed cultural facility in July 2019 and even higher than in September 2019. However, noticing an upward trend in tourist traffic in 2018 and 2019, it can be assumed that the possible sale of tickets in the summer season in 2020 could be 76,000 in July, 95,000 in August, and 53,000 in

September. In terms of the total loss in ticket sales share in the summer period (without the division into reduced tickets and regular tickets), Książ Castle was potentially visited by over 23% fewer tourists.

The second cultural object analyzed by the authors of this study is the Kuks complex. The tourist offers are slightly different from that of Książ Castle, as tourists can visit a significant part of the object free of charge. However, seeing some of the buildings' interiors is possible after purchasing tickets, which can be purchased at a machine. After several years of reconstruction, the Castle was opened to tourists in the last days of March 2019, and by the end of November 2019, around 140,000 tourists had purchased tickets. Converting to the number of tickets sold per month, this is the value of over 1550 tickets. The authors assumed a similar trend for the Kuks complex by analyzing the intensity of tourist traffic throughout the year in Książ Castle in 2001–2016 and the data in Table 3. Both objects are cultural objects located in a relative geographic and climatic location. Table 3 below presents the estimated percentage of tourist traffic in Książ Castle each month from 2001 to 2016.

**Table 3.** The estimated percentage of tourist traffic in Książ Castle each month from 2001 to 2016.

| January | February | March | April | May | June | July | August | September | October | November | December |
|---------|----------|-------|-------|-----|------|------|--------|----------|---------|----------|----------|
| 2.3 | 2.5 | 2.5 | 6.4 | 14 | 8.4 | 17 | 19 | 15 | 8.4 | 2.5 | 2 |

Source: own study based on data obtained from the Książ Castle Enterprise and [70].

Using the obtained monthly percentage shares for tourist traffic in Książ Castle, the authors estimated that the tourist traffic in the Kuks baroque complex in the summer months of 2019 in total (people purchasing a ticket) amounts to approximately 77,000 people. Additionally, according to the Kuks Information Center data, it is estimated that about 40% of visitors to the Kuks complex visit the facility without purchasing a ticket. Therefore, it can be assumed that the total tourist traffic during the summer holidays in 2019 was at the level of 108,000 people. This number makes Kuks one of the most visited tourist destinations in the Hradec Králové region. In comparison, the Książ Castle, in the 2019 summer period, amounted to tourist traffic of 218,019 people in total. In 2020, during the pandemic, the complex of baroque buildings in Kuks' village was closed in March, May, October, November, and December. In 2020, during the complex's opening, 59,000 tickets were sold (within seven months). In 2020, the involved managers did not keep monthly statistical data, and all tickets were sold at a discount. Assuming that the volume of tourist traffic in the Kuks baroque complex could be comparable to the tourist traffic in Książ Castle during the COVID-19 pandemic, it can be concluded that in 2020 in the summer period, the number of tourists visiting the facility was over 64,000 people.

Table 4 below presents the estimated intensity of tourist traffic in the Kuks baroque complex in the last two years in the summer period, in 2019 and 2020.

**Table 4.** Shows the number of tourists visiting cultural sites in the summer of 2019–2020.

| The Number of Visitors to Kuks | July | August | September |
|--------------------------------|------|--------|-----------|
| 2019 | 35,900 | 40,173 | 31,715 |
| 2020 | 21,340 | 23,851 | 18,830 |

Source: own study based on data obtained from the Kuks cultural buildings.

Table 3 above shows that the Czech Republic's tourist traffic in 2020 decreased by over 40% compared to the previous year. It is undoubtedly the effect of the ongoing COVID-19 pandemic, which is confirmed by the data on the number of discount tickets for seniors at Książ Field and information from the Kuks Information Center. The list of ticket sales with the division into reduced tickets for children and youth in education, regular tickets, and tickets for seniors, i.e., people over 65 years of age in Książ Castle, is presented in Figure 2 below.

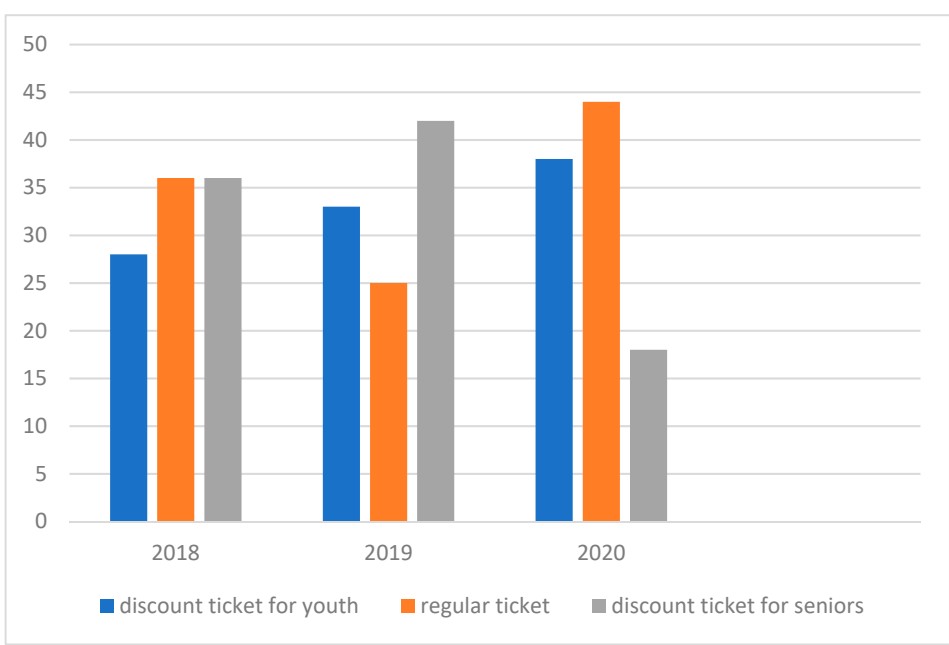

**Figure 2.** Ticket sales at Książ Castle during the summer season in 2018–2020. Source: own study based on the Książ Castle data.

Figure 2 above shows the percentage share of ticket sales in Książ Castle, broken down by age. The first age groups are children and adolescents who have student IDs. Thus, the upper age limit for the first group is 26 years. Another group of people is professionally active people aged up to 65. The last group is the so-called seniors who have a retirement card, thanks to which they can also purchase reduced tickets. The senior age group in 2018 and 2019 was characterized by a significant share in the sale of reduced tickets. The recorded more significant tourist traffic in this age group resulted from many visiting seniors from other countries. The following year, in 2020, the share in the sale of tickets for seniors decreased by only 18%, compared to 42% of the claimed ticket sales in 2019. Less tourist traffic of people at a high risk of developing COVID-19 is entirely understandable and predictable. However, in the current (2020) year, the share of economically active people, i.e., in the 27–64 age group, has increased.

## 5. Value Propositions for the Client during the Pandemic Introduced by the Managers at Książ Castle and the Kuks Complex

The growing interest in the audio guide service among people visiting cultural sites from the perspective of those managing the Książ Castle is a desirable trend. This situation is because the audio guide offering generates more revenue for the facilities than the guide service.

The guides at Książ Castle are hired under a mandate contract, and for one session, they receive a gross remuneration of PLN 30. The estimated time of the visit is 1 h. The cost for visitors to Książ Castle with the purchased guide service is PLN 45 for a regular ticket and PLN 38 for a reduced ticket. The minimum number of people to buy this service is 15, and 35 people is the maximum number. The Castle management cooperates with fourteen guides. For organized tours, it is necessary to book such a service at least 2 h in advance. For people who spontaneously form the required group size to buy a guide, there must be two guides in the facility, which also have a remuneration of PLN 15/hour for readiness to work.

From the visitor's perspective, the audio guide service is cheaper, as a regular ticket costs PLN 39, and a reduced ticket costs PLN 32 per person. The device rental time is up to 90 min. The Castle has 600 devices. During a pandemic, 200 devices may be made available at any time. After their use, they must be disinfected. With this solution, it is not possible

to reserve devices in advance. The cost of buying an audio guide device is about PLN 320 gross per piece.

The information obtained from Książ Castle shows that during the holiday season, 10% of visitors used the guide service and 55% the audio guide service; the remaining 35% was the sale of tickets for individual visits. In the survey, approximately 65% of respondents declared that they had purchased the audio guide service, i.e., out of 212 respondents, 138 people bought the audio guide service, 2.5% of the respondents purchased the guide service, and 32.5% only purchased a ticket.

For simplicity of calculation, the declarations of respondents in Książ Castle, who purchased the audio guide and guide service at a discounted ticket price, were used. For this small number of people, it can be calculated that the company's revenue was PLN 114 (sale of tickets with the guide service) and PLN 4416 (ticket sales). Table 5 below presents the data obtained from the Książ Castle on the number of tourists visiting in the summer season in 2020 and the share in ticket sales, broken down into a guide service and an audio guide obtained from the surveys and interviews conducted below.

**Table 5.** Comparison of the revenue for guide and audio guide services at Książ Castle in the summer of 2020.

|  | July | August | September | Sum |
|---|---|---|---|---|
| number of tourists visiting Książ Castle | 58,000 | 71,000 | 37,000 | 166,000 |
| 10% of tickets with guide service | 5800 | 7100 | 3700 | 16,600 |
| 65% of tickets with the audio guide service | 37,700 | 46,150 | 24,050 | 107,900 |
| Income from the purchased discount tickets with the guide service | 220,400 | 269,800 | 140,600 | 630,800 |
| Revenue from the purchase of discount tickets with the audio guide service | 1,206,400 | 1,476,800 | 769,600 | 3,452,800 |

Source: own study based on the Książ Castle data and surveys.

Table 5 above shows that the audio guide service for people managing Książ Castle is a more advantageous offer. It generates more income. It constitutes 84.6% of the share of the guide's service offer, which was 15.44%. The obtained results confirm the data presented in Figure 2 regarding the percentage share of ticket sales with the guide service and audio guide service during the holiday season in 2020. In previous years, however, the trend was reversed, and the guide service had a more significant share in ticket sales.

This article's authors conducted a similar simulation for the complex of baroque palaces in Kuks in the Czech Republic. From the surveys and interviews conducted, a total of 132 people were included in the results. A total of 29% of respondents said they would like to buy tickets for the guide service. On the other hand, 63% expressed their will to purchase an audio guide service. Among this group, 27 people decided not to buy an individual sightseeing ticket but declared that if the facility had an audio guide service, they would buy such a ticket. The Kuks building complex's potential revenues during the pandemic are shown in Table 6. The revenue potential was based on the ticket purchase price for the Kuks facilities, amounting to PLN 23. Currently, the facility does not have an audio guide service, so the authors of the article consider the proportions in the price of tickets with the audio guide service and the guide service offered at Książ Castle, using the same ratio of 16%.

As in the case of Książ Castle, there is a more significant revenue potential for the facility in the Czech baroque complex Kuks if an audio guide is available. For people who participated in the survey and purchased tickets for a total amount of PLN 1771, if the center had an audio guide service, the revenue would be higher by about PLN 2464, i.e., about 16% more. While maintaining the potential for revenue growth of around 16%, if the tourist destination has the audio guide service of the Kuks palace facility manager, they could expect such an increase in revenue.

**Table 6.** Comparison of the potential revenue for guide and audio guide services in the Kuks baroque complex in the Czech Republic during the summer of 2020.

|  | July | August | September | Sum |
|---|---|---|---|---|
| number of tourists visiting Kuks | 21,340 | 23,851 | 18,830 | 64,021 |
| 29% of tickets with guide service | 61,886 | 691,679 | 54,607 | 1,856,609 |
| 63% of tickets with the audio guide service | 134,442 | 1,502,613 | 118,629 | 4,033,323 |
| Potential income from the purchased discount tickets with the guide service | 1,423,378 | 1,590,862 | 1,255,961 | 4,270,201 |
| Potential income from the purchased discount tickets with the audio guide service | 3,629,934 | 4,057,055 | 3,202,983 | 1,088,997 |

Source: own study based on Kuks Information Center data and surveys.

## 6. Conclusions

Answering the research questions posed in the Section 2, the authors identified universal recommendations for cultural areas, which can be used in times of unforeseen crises and random events and can also respond to permanent changes in consumer choices (Table 7) [71].

**Table 7.** Research questions and conclusions.

| Research Questions | Książ | Kuks |
|---|---|---|
| How did the level of visitors change with the pandemic at the two sites? | Decreased by 23%. | Decreased by over 40%. |
| How have changes in the level of ticket sales affected the overall operation of the two facilities? | E-marketing and e-sales activities have increased. | Reduced activity in promoting the facility. |
| How was both facilities' adaptation and response process during the pandemic? | Solutions to increase the safety of tourists were introduced very quickly. | After a few months, adjustment measures were introduced to improve the safety of tourists. |
| What measures have been taken at both facilities to maintain financial stability in connection with the pandemic? | Reduced revenues have halted investment and renovation activities. | The number of staff has been reduced, and current costs have been significantly reduced. |
| What actions have been taken by both facilities to change competitive instruments (product, price, distribution channels, promotional activities)? | New products have been introduced that are linked to changes in price and distribution, such as a personalized tour with audio guides. | The possibility of virtual tours in place of traditional tours has been developed. |
| How did both facilities react to their customers during the pandemic? With particular attention to solutions relating to the green economy, what resources did both facilities tap into? | Revitalization of outdoor gardens and thermal upgrades inside the building. Thanks to this, the facility can apply to be a health resort. | Thermal modernization of the building's heating system. In the future, reduce heating costs. |
| What lasting changes in buying behavior and expectations did the pandemic cause among customers of historic sites? | Increased the share of individual tours without a guide. | Reduction in the number of tourists visiting directly by eliminating visits to the main parts of the Castle, expanding the number of virtual tourists. |
| What added value did both venues introduce during the pandemic, and what added value did their customers expect? | Introduced: individual tours Expected: change in ticket prices | Introduced: virtual tours Expected: opportunities for safe tours |
| What recommendations can be defined for similar facilities in both countries at the level of such crises emerging? Additionally, which recommendations can be adopted permanently as a continuous improvement in their services? | After a decline in the months of III-V, there was a sharp increase in June and July 2020, then stability was observed. | The decline was pronounced until autumn 2020; restoration of visitor counts occurred from spring 2021. |

Source: own study based on Kuks Information Center data, Książ documents, and surveys [72].

The number of visitors during the coronavirus pandemic to the analyzed tourist facilities has significantly decreased, especially when people at high risk of a severe disease course and possible complications have increased. Government amendment of significant restrictions in terms of movement and the possibility of visiting tourist facilities contributed to the fact that this branch of the economy's revenues in 2020 decreased drastically. According to a survey by the Statistical Office, 86,400 tourists received 326,200 accommodations. It was lower than April last year by 96.5% and 94.5%, respectively! A decrease in the number of tourists was also recorded in May 2020. According to estimates, the number of people having overnight stays compared to the same month last year was lower by approximately 88.1%. Restrictions on the use of tourist facilities also impact the decline in people visiting cultural sites. Thus, it creates a field for organizing so-called Green tourism, which, as a green initiative, will support the development of tourism in general. The above data are the result of the direct impact of the pandemic on society and the economy. In addition, there were also indirect effects of the pandemic, namely new trends in consumer choices, including increased interest in the audio guide service in tourist facilities. The analyses carried out in 2020 at Książ Castle show that the audio guide service had a more significant share in the facility's revenues than the guide service due to the increased interest. In addition, the surveys and interviews show that those who wanted to explore the mystery of the history of the walls of the building and at the same time maintain high sanitary standards decided to purchase tickets with an audio guide service. After the end of the tour, each person positively assessed the service and even declared that it would be possible to use it in other facilities in the future—about 98% of respondents in Książ Castle did so. The Kuks baroque palace complex does not offer an audio guide, and among the answers received, over half of the people who visited the facility for free tours expressed their interest in the offer and readiness to buy a ticket for such an option.

With the collected data, the authors of this study provide a positive answer to the question posed in the introduction to this article on the changes in society's needs regarding the manner of visiting cultural sites. The growing interest in the audio guide service was also noticeable in 2019, while the pandemic accelerated these changes. From the perspective of people managing the facilities, this is a desirable trend, as it generates more revenue for the facilities. However, will the change in the structure of tourism also be a constant trend? It is difficult to answer this question. Indeed, fear among people at increased risk of severe disease from SARS-CoV-2 will slow down the tourism activity of this group of society, but with time, when Europe takes control of the spread of the virus, it can be assumed that the structure and tourism dynamics will slowly resemble the one before the pandemic. High uncertainty resulting from the course and duration of the pandemic, even over a two-year perspective, significantly limits the possibility of taking long-term action. However, tourism policy entities will likely undertake future initiatives to help tourism economy entities and their employees over a more extended time. After the pandemic is over, the situation in the tourism market will stabilize to the pre-pandemic state.

Moreover, there is a need to develop a system of actions that can be used in potential future crises. The scope of the public policy concerning economic entities, aimed at counteracting and limiting the adverse effects caused by the sanitary regime, consists of general actions aimed at all financial entities and specific activities covering the tourism industry. The available measures include loan guarantees, co-financing employee salaries, cancellation or extension of tax liabilities, and reductions or deferrals of social security contributions. Based on the analysis of the document prepared by the Polish Tourist Organization, presenting an overview of solutions used as part of the tourism policy in 18 selected countries of the world that have been strongly affected by the epidemic crisis, tourism policies aimed at the following:

- Tourism economy entities: loan programs dedicated to tourism enterprises, co-financing salaries of employees of tourism enterprises in the period of lack of revenue;
- Tourism demand: promoting and encouraging domestic tourism, based on the so-called green initiative, i.e., promoting tourism in relations with neighboring countries

with a similar level of epidemic risk; promoting under tourism, the tourism to places and areas with a low level of tourist penetration; support for family tourism, individual and small group travel; promotion of slow tourism; support for personal transport; and conducting an information policy in the field of safety, health, and hygiene. It can therefore be concluded that the "green economy" is becoming an essential instrument for overcoming a crisis or economic downturn, as well as for fighting unemployment;

- Both the supply and demand side of the tourism market: use of vouchers in place of travel offers purchased earlier and not realized due to epidemic restrictions (airlines, travel agencies), with a deferred use date; thus, the state limits the possibility of reimbursement of expenses incurred by clients to tourist enterprises if the services have not been provided; the use of vouchers for services that will be purchased and provided in the future; and the introduction of travel vouchers, financed or co-financed by the state and entities employing employees through which it will be possible to pay for tourist benefits, usually within the territory of the country. All the activities mentioned above at the current stage of the pandemic development (May 2020) relate to the short-term action horizon. Long-term solutions are not yet planned. It should be expected that on the European Union scale, long-term activities will be covered by the following financial perspective for the years 2021–2027;

- Although the economic benefits of tourism are undeniable, it also has a significant impact on the environment. Environmental degradation, climate change, and the pandemic's timing significantly impact overall tourism trends. Moreover, changing weather trends can destroy what attracts tourists. It is also essential that more and more tourists aware of environmental issues are looking for "green destinations" that respond to the challenges related to, among other things, environmental protection; e.g., tourists take into account the availability of "green services" when booking. Tourism and environmental sustainability are becoming natural partners with increasingly interconnected action plans.

The added value that the authors observed, as actions introduced or desirable in both historic sites, not only during the pandemic, are as follows: 1. increasing safety in the area of personal hygiene and human contact through the introduction of tools such as the development of special handling instructions; 2. paying attention to green management, which results from savings and customer expectations; 3. using tools from the area of digitization both when selling services and enriching the product, distribution, and promotion of services in the two historic sites.

According to obtained added value, the authors consider that one of this study's main conclusions is using and developing artificial intelligence (AI), which would enable a virtual tour of the monuments. Such offers have been made in hotels but are not being created on cultural sites. The virtual tour also corresponds with the need for personal hygiene. Such paid offers could be an answer for people who cannot visit the sites not only because of pandemics but also due to individual conditions. Such offers could also be directed to distant schools as teaching materials. Another universal solution is creating a multicultural social portal where managers of cultural centers could exchange good practices and adapt their needs along with the current expectations of their clients. It would be an excellent exchange of information and demands.

**Author Contributions:** Individual contributions, conceptualization: M.P.—40%; M.R.—30%; J.T.—30%; methodology: M.P.—30%; M.R.—30%; J.T.—40%; validation: M.P.—35%; M.R.—35%; J.T.—30%; formal analysis: M.P.—40%; M.R.—30%; J.T.—30%; resources: M.P.—25%; M.R.—35%; J.T.—30%; writing—original draft preparation: M.P.—40%; M.R.—35%; J.T.—30%; writing—review and editing: M.P.—40%; M.R.—30%; J.T.—30%; visualization: M.P.—70%; J.T.—30%; supervision: M.P.—35%; M.R.—35%; J.T.—30%; funding acquisition: M.P.—33%; M.R.—34%; J.T.—33%. All authors have read and agreed to the published version of the manuscript.

**Funding:** This research received no external funding.

**Institutional Review Board Statement:** Not applicable.



**Informed Consent Statement:** Informed consent was obtained from all subjects involved in the study.

**Data Availability Statement:** Data sharing not applicable.

**Conflicts of Interest:** The authors declare no conflict of interest.

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
