# Peer review of "Added Value on a Day in the Pandemic in Tourist Attractions in the Polish–Czech Borderland as a Green Economy Initiative"

_sustainability, doi:10.3390/su15042911_

Round 1
Reviewer 1 Report
The research is interesting for the journal audience, and follows the philosophy of the journal. Although the paper is interesting, it is unsatisfactory from several points of view.
The paper introduces too many concepts without defining them and reviewing previous literature related to it. When a research problem is formulated, there needs to be a thorough literature review of previous research. It is not elaborated enough here. The paper needs to explain which concepts that has been addressed and why. It is also prerequisite to include the review itself. Based on the review one should highlight the gap in the literature and develop the research problem and questions.
Introduction
Actually the paper does not include Introduction and theoretical grounding. I do not understand it as the Authors reported the text as an article.
The content of the paper starts with Methodology of own research. However, first the article should contain theoretical grounding of the problem referred in the empirical study. Second, in my opinion the content of this section is rather a kind of introduction, not methodology. I invite the author to revise the introduction to write about the motivation for this research. Thus, in the Introduction, you should say what kind of use you envisage for the study results. Are they supposed to provide information for tourist attractions companies? Are they supposed to help the tourist attractions to strengthen their attractiveness or so?
Theoretical grounding
In general paper lacks theoretical grounding. The Authors provide some theory (in the first section). However, it is needed to explain which concepts that has been addressed in the study and why. For example, on page 7 the Authors write about value proposition for Książ Castle and Kuks Complex but they do not mention the concept of value proposition in the theoretical grounding. The same refers to 4A concept (page 4). There are no references to this theory. What about 6 A’s model for tourism attractiveness. The Authors do not mention it at all. In general, the part of the paper regarding the concept of ‘customer value’ is quite chaotic and it definitely lacks relevant reffferences.
On page 9 the Authors write about providing the answer for the question posed in the introduction. However, this question is not clear at all. I suggest to supplement the paper with more clear 2-3 research questions. This would provide a kind of a roadmap for the reader.
Methodology
In abstract the Authors write about ‘questionnaire method’. However, the is no section regarding the procedure of the study, explain conducting the survey. It should be added before presenting the findings. As reading the text of the paper, I would rather say that the Authors applied here is multiple case study (Zamek Książ and Kuks Complex) methodology including analysis of documents and interviews. But there is no information neither about interview (survey) nor about the research participants. Simply, there is no information about the study method.
Discussion
The paper does not include any discussion. Discussion section is crucial for academic papers and should be about contrasting the manuscript's findings with the existing literature. Please supplement the paper with Discussion section containing the information to what extent your results are aligned with the evidence found in the adequate literature. To what extent your results are not aligned? Please provide a more comprehensive discussion on this.
Conclusions
Conclusions section should provide answers for the research questions posed by the Authors. It also emphasize the study contribution to the current theory and research on the analysed topic as well as emphasize authors’ recommendations for researchers and practitioners based on their research findings. Moreover, Conclusions section should be supplemented with the paragraph regarding the limitations of the conducted study and some further research avenues referring to the analyzed topic. Please, revise this section and add all needed paragraphes.
Author Response
Response to Reviewer Comments
In the beginning, the authors of the following article would like to thank you very much for your valuable comments and tips. Their inclusion undoubtedly raised the quality of the paper significantly. Below are the main changes made.
Point 1: The paper introduces too many concepts without defining them and reviewing previous literature related to it. When a research problem is formulated, there needs to be a thorough literature review of previous research. It is not elaborated enough here. The paper needs to explain which concepts that has been addressed and why. It is also prerequisite to include the review itself. Based on the review one should highlight the gap in the literature and develop the research problem and questions.
Response 1: In the work, the concepts addressed were revised along with their literature translation. For this purpose, new sections were added to the work ( as recommended by the reviewer). Table 1 Definitions of the term "Consumer Behaviour" was added. An extensive literature review was conducted, and the article was enriched with 36 new literature items. The gap in the literature shows the area of non-understanding of the impact of Artificial Intelligence (AI), including Machine Learning (ML) and Digital Marketing (DM), on modern sales (Lilien, 2016; Paschen et al., 2020a). On the other hand, the research gap is to identify those added values for clients of historical sites that should be introduced regardless of the pandemic period due to permanent changes in client behavior and expectations.
Point 2: Actually the paper does not include Introduction and theoretical grounding. I do not understand it as the Authors reported the text as an article.
The content of the paper starts with Methodology of own research. However, first the article should contain theoretical grounding of the problem referred in the empirical study. Second, in my opinion the content of this section is rather a kind of introduction, not methodology. I invite the author to revise the introduction to write about the motivation for this research. Thus, in the Introduction, you should say what kind of use you envisage for the study results. Are they supposed to provide information for tourist attractions companies? Are they supposed to help the tourist attractions to strengthen their attractiveness or so?
Response 2: According to the comment, the authors revised the Introdution and changed the article's main sections ( 1. Introduction, 2. Material and Methods, 3. Discussion, 4. Analysis of the market situation of both studied objects in the period preceding the pandemic and during the pandemic, 5. Value propositions for the client during the pandemic introduced by the managers at Książ Castle and the Kuks Complex, 6. Conclusions.
Introduction. We refilled it with information about why the topic is essential and what is the research gap. We also presented the added values, which might be the future way for the touristic branch.
"The topic is important because the situation during the pandemic showed the lack of a quick response, which is only possible if you have prepared scenarios for the crisis. Modern times are often defined as a time of permanent crisis. Therefore, it is crucial for those running companies today, including heritage managers, to constantly observe how to improve and create new added value to retain and/or acquire customers. On the other hand, the research gap is to identify those added values for clients of historical sites that should be introduced regardless of the pandemic period due to permanent changes in client behavior and expectations. The added values that the authors observed, as activities introduced or desired at both historic sites not only during the pandemic, are:
- Increasing the safety of personal hygiene and interpersonal contacts through the introduction of tools such as the development of a particular instruction manual;
- Paying attention to green management, which resulted from savings, as well as customer expectations.
- Using tools from the area of digitization both when selling services and enriching the product, distribution, and promotion of services in the two historic sites.
In the coronavirus era, the consumer's attitude has changed because his activities related to market activity were influenced by a different way of thinking and assessing the situation. The current lifestyle, focusing on the problems of frequency of shopping, participation in social circles, entertainment, recreation, hobbies, or the very attitude to work, had to change, regardless of the will of a given person. This surprising situation changed customers' personalities, which, as it was found, under stable conditions, changes over the years, and today's time has accelerated and changed us, the environment, and even our life goals, plans, and priorities. Marketing changes behaviors, habits, views of individuals and social groups, and the value system under the influence of specific activities, called marketing activities. Practical marketing activities are now considered to have a real, measurable impact on the financial result, profitability, or value of the enterprise, especially by increasing the total customer value (Woźniczka et al., 2014, p. 18). New marketing trends are reflected in social, cultural, and ecological aspects resulting from buyer behavior changes, which new organizations should remember. As shown by market laws, enterprises cannot only focus on business goals but must also consider the principles of the green economy.
Point 3. Theoretical grounding. In general paper lacks theoretical grounding. The Authors provide some theory (in the first section). However, it is necessary to explain which concepts that has been addressed in the study and why. For example, on page 7 the Authors write about the value proposition for Książ Castle and Kuks Complex but they do not mention the concept of value proposition in the theoretical grounding. The same refers to 4A concept (page 4). There are no references to this theory. What about 6 A's model for tourism attractiveness. The Authors do not mention it at all. In general, the part of the paper regarding the concept of 'customer value' is quite chaotic and it definitely lacks relevant reffferences. On page 9 the Authors write about providing the answer for the question posed in the introduction. However, this question is not clear at all. I suggest to supplement the paper with more clear 2-3 research questions. This would provide a kind of a roadmap for the reader.
Response 2: To better organize our paper, we changed the section Methodology into Material and Methodology, where we added references to the theory. We used more than 29 new literature positions to make it more valuable for the readers. Below are examples of a few literature positions used in the paper.
- Higgins-Desbiolles, F. (2020). Socialising tourism for social and ecological justice after COVID-19. Tourism Geographies, 22(3), 610-623. https://www.tandfonline.com/doi/full/10.1080/14616688.2020.1757748.
- Crossley, E. (2020). Ecological grief generates desire for environmental healing in tourism after COVID-19. Tourism Geographies, 22(3), 536-546. https://doi.org/10.1080/14616688.2020.1759133.
- Niewiadomski, P. (2020). COVID-19: from temporary de-globalization to a rediscovery of tourism? Tourism Geographies, 22(3), 651-656. https://doi.org/10.1080/1 4616688.2020.1757749.
- Borodako, K. (red.) (2021). Turystyka w okresie pandemii. Bogucki Wydawnictwo Naukowe, https://doi. org/10.12657/9788379863501-1.
- Paschen, J., Paschen, U., Pala, E. and Kietzmann, J. (2020a), Artificial Intelligence (AI) and value co-creation in B2B sales: activities, actors and resources. Australasian Marketing Journal, https://doi.org/10.1016/j.ausmj.2020.06.004.
- Syam, N. i Sharma, A. (2017). Waiting for a sales renaissance in the fourth industrial revolution: Machine learning and artificial intelligence in sales research and practice. Industrial Marketing Management, 69, 135–146. https://doi.org/10.1016/j.indmarman.2017.12.019.
- Lilien, G. L. (2016). The B2B knowledge gap. International Journal of Research in Marketing, 33(3), 543–556, https://doi.org/10.1016/j.ijresmar.2016.01.003.
- Panagopoulos, N. G. I Avlonitis, G. J. (2010). Performance implications of sales strategy: The moderating effects of leadership and environment. International Journal of Research in Marketing, 27(1), 46–57. https://doi.org/10.1016/j.ijresmar.2009.11.001.
- Johnson, J. S. (2015). Qualitative sales research: an exposition of grounded theory. Journal of Personal Selling & Sales Management, 35(3), 262–273. https://doi.org/10.1080/08853134.2014.954581.
- Bothe, J., Weiss, P. and Dumitrescu, B. I. (2021). Success factors for digital sales development in B2B sales of products requiring explanation. W V. Pamfilie, L. Dinu, D. T chiciu, C. Ple ea i C. Vasiliu (red.). 7th BASIQ International Conference. New Trends in Sustainable Business and Consumption. https://www.researchgate.net/publication/352132599.
- Murawska, M. (2020). Zmiany indeksów gieÅ‚dowych w okresie bessy wywoÅ‚anej pandemiÄ… COVID-19 w pierwszym kwartale 2020 r. Nowoczesne Systemy ZarzÄ…dzania, 15(4), pp. 79–93, https://doi.org/10.37055/nsz/134106.
- Geiger, S. i Guenzi, P. (2009). The sales function in the twenty-first century: where are we and where do we go from here? European Journal of Marketing, 43(7/8).
In this section, we underlined the six research problems and eight research questions, hoping it will provide a kind of roadmap for the reader.
(p. 8) The research problems focused on the following:
- Determining the quantitative relationship between pandemic-related insults at different times of the pandemic and the level of ticket sales;
- Identifying the response of businesses to changes resulting from the pandemic in general and concerning green economy solutions;
- Identifying the level of customer satisfaction with the changes made;
- Identifying the added value accepted and postulated by customers of both monuments;
- Identifying differences in the practice of the two monuments in two different countries during the pandemic;
- Identifying the added value for the customer in both monuments. As a recommendation for other such facilities in Poland and the Czech Republic concerning the green economy.
In connection with the development of the problem areas, the authors identified the following research questions:
- How did the level of visitors change with the pandemic at the two sites?
- How have changes in the level of ticket sales affected the overall operation of the two facilities?
- How was both facilities' adaptation and response process during the pandemic?
- What measures have been taken at both facilities to maintain financial stability in connection with the pandemic? What actions have been taken by both facilities to change competitive instruments (product, price, distribution channels, promotional activities)?
- How did both facilities react to their customers during the pandemic? With particular attention to solutions relating to the green economy, what resources did both facilities tap into?
- What lasting changes in buying behavior and expectations did the pandemic cause among customers of historic sites?
- What added value did both venues introduce during the pandemic, and what added value did their customers expect?
- What recommendations can be defined for similar facilities in both countries at the level of such crises emerging? And which recommendations can be adopted permanently as a continuous improvement of their services?
Point 4. Methodology In abstract the Authors write about 'questionnaire method'. However, the is no section regarding the procedure of the study, explain conducting the survey. It should be added before presenting the findings. As reading the text of the paper, I would rather say that the Authors applied here is multiple case study (Zamek Książ and Kuks Complex) methodology including analysis of documents and interviews. But there is no information neither about interview (survey) nor about the research participants. Simply, there is no information about the study method.
Response 4: To clarify what methods the authors had used in their study, we added a sentence: (p.8). The main objective of the conducted research was to identify the added value that was generated or should be implemented by companies managing historical sites in both Poland and the Czech Republic - during the pandemic based on: internal documents of both organizations, analyses of the level of tourist traffic, interviews with managers and surveys among tourists, which both companies carry out on an ongoing basis. The authors also conducted their research through face-to-face interviews among tourists at both facilities. The questionnaire took 132 respondents between June and August 2020 at both facilities
During the study, the authors used the following research methods: simple statistical methods, comparative analysis, analysis of foundational documents in both entities, interviews with the management of both facilities, and discussion and analysis of surveys conducted.
Point 5 Discussion The paper does not include any discussion. Discussion section is crucial for academic papers and should be about contrasting the manuscript's findings with the existing literature. Please supplement the paper with Discussion section containing the information to what extent your results are aligned with the evidence found in the adequate literature. To what extent your results are not aligned? Please provide a more comprehensive discussion on this.
Response 5: Following the reviewer's valuable comment discussion section has been added:
(pp.9-10) Most of the available research results on consumer behavior resulting from the onset of a pandemic concern the first few months after the outbreak. They cover selected aspects of purchasing behavior in various product categories. The results of studies conducted in the early period of the pandemic indicated that consumers changed their behavior and sharply increased their purchases of certain products (Aydınlıoğlu, Gencer, 2020; Islam et al., 2020; Laato et al., 2020). As M. Loxton et al. (2020) point out, consumers in the first months of the epidemic behaved similarly to previous crises, such as during the 2007-2009 financial crisis - they bought products in a panic, reduced spending on elective goods and made their decisions based on media reports
The COVID-19 pandemic has significantly affected tourism. Many places that have enjoyed enduring popularity for years saw a significant drop in visitors in 2020. The introduction of restrictions in the form of a ban on movement has contributed to a substantial decline in tourist arrivals in Poland and the Czech Republic. The literature describes the changes that COVID-19 has caused in the tourism economy (Goodgrer and Kieran, 2020), formulates multivariate forecasts of the development of the situation, draws scenarios for recovery from the crisis, and proposes corrective measures. A special issue of Tourism Geographies (vol. 22, no. 3) was published, containing as many as 25 articles on COVID-19-related topics, for example, outlining a vision for post-emergency tourism (Haywood, 2020), proposing necessary economic measures to save the tourism economy (Cave and Dredge, 2020), and pointing to the required transformation (Hall, Scott, and Gössling, 2020). There are theses about the need to socialize and green tourism after COVID-19 (Higgins-Desbiolles, 2020; Crossley, 2020) with a strong emphasis on sustainability and responsible tourism (Niewiadomski, 2020). It has been proven that during recovery from an epidemic crisis, traveling within one's own country and staying at agritourism farms and facilities to ensure sanitary safety will become more popular (Borodako, 2021). In the era of pandemics, business models in tourism, in the area of sights, referred to the opportunities offered by the virtual world. We are seeing more and more discussions related to changes in the progressive digitization of sales processes (Paschen et al., 2020b; Syam and Sharma, 2017). Researchers recognize a gap in the area of understanding the impact of Artificial Intelligence (AI), including Machine Learning (ML) and Digital Marketing (DM), on modern sales (Lilien, 2016; Paschen et al., 2020a). The literature contains numerous studies on the distinction between marketing and sales (Panagopoulos and Avlonitis, 2010).
In contrast, research on how consumer behavior changes affect the sales function's remodeling relative to the marketing function is an episodic phenomenon (Johnson, 2015). Currently, the biggest challenge facing businesses, management trends 62 e-mentor No. 4 (91), is the COVID-19 pandemic, which affects all business areas, including the sales function (Bothe et al., 2021). A pandemic outbreak is part of a broader problem of various phenomena referred to as exogenous shocks, which tend to have a jumping effect on changes in economic and social processes (Murawska, 2020). They are part of the peculiarities of standard economic shocks - demand, supply, price, and financial shocks, which have affected almost all industries, forcing companies to reformulate their implemented strategies, including sales strategies (Geiger and Guenzi, 2009; Panagopoulos and Avlonitis, 2010; Radlinska, 2020; Rangarajan et al., 2018). Knowledge in this area is drawn mainly from commercial publications, consulting firms' reports, and practitioners' studies (Lane and Piercy, 2009; Leigh and Marshall, 2001; Palmer and Flanagan, 2016). Overall, during the COVID-19 pandemic, a theory can be advanced that the tourism system is more vulnerable than other systems [Espiner et al. 2017]. The pandemic caused unprecedented effects: it affected tourists' lifestyles, behaviors, and travel patterns [Wassler, Fan 2021]. In addition, an overall increase in mental disorders caused by isolation, such as increased anxiety, impacted the frequency and form of tourism participation [Ahmed et al. 2020; Kock et al. 2020]. The pandemic also resulted in other adverse consequences, such as creating a negative image of travel [Godovykh, Ridderstaat 2020]. Quite a bit of change could also be seen in the destination's vision, which, as is well known, can change constantly. The COVID-19 pandemic significantly impacted its further formation [Zenker, Kock 2020]. Overall, it affected global supply and demand [Abu Bakar, Rosbi, 2020]. Hence, it is understandable that there has been a marked increase in interest in the issue of tourism crisis and crisis management [Sigala 2020; Baum, Hai 2020; Hall et al. 2020], among others, in the context of pandemic mitigation and post-crisis recovery of the industry [Yeh 2021].
In the Polish literature, one can find studies of a pilot and non-representative nature (Gorzelany--Dziadkowiec, 2020; Samuk, Sidorowicz, 2020). However, they allow us to see symptoms of changes in the structure of spending (reduction of clothing and footwear purchases), increased online shopping activity, and the use of instant messaging in the purchasing process. A change in consumer attitudes has been identified: the growing importance of rest, relaxation, and a healthy lifestyle. In the pages of the journal Organization Review, the results of the study of consumer behavior in light of COVID-19 have not yet been published, but the issues of sustainability of behavioral changes under the influence of modification of marketing activities of enterprises and habits of buyers have been considered (Brzezinski, 2020). The author of the cited article put forward the thesis that the changes observed during the pandemic will be permanent.
When considering the tourist attractions of the Polish-Czech border region as a green economy initiative, it is necessary to point out what the green economy is. The concept has emerged in debates about sustainable development, especially in light of the Rio+20 idea [United Nations Conference on Environment and Development meeting in Rio de Janeiro from June 3 to 14, 1992]. For the aims of this article, it has been assumed that the green economy generally refers to an economy in which the quality of human life and the state of the environment are considered broadly when making production and consumption decisions. Hence, the inclusion of tourist attractions of the Polish-Czech borderland in this concept/initiative should be considered a novelty.posiedzeniu w Rio de Janeiro w dniach od 3 do 14 czerwca 1992 r.].
Point 6 Conclusions Conclusions section should provide answers for the research questions posed by the Authors. It also emphasize the study contribution to the current theory and research on the analysed topic as well as emphasize authors' recommendations for researchers and practitioners based on their research findings. Moreover, Conclusions section should be supplemented with the paragraph regarding the limitations of the conducted study and some further research avenues referring to the analyzed topic. Please, revise this section and add all needed paragraphes.
Response 6: Conducted in both the literature and own research in the form of face-to-face interviews with the managers of the analyzed monuments, as well as obtained answers in a survey conducted among visitors to the monuments, the authors formulated answers to the research questions posed in the Material and Methods section. The added value that the authors observed, as actions introduced or desirable in both historic sites not only during the pandemic, are: 1. Increasing safety in the area of personal hygiene and human contact through the introduction of tools such as the development of special handling instructions; 2. They are paying attention to green management, which results from savings and customer expectations. 3. using tools from the area of digitization both when selling. According to obtained added value, the authors consider that one of the study's main conclusions is using and developing artificial intelligence (AI), which would enable a virtual tour of the monuments. Such offers have been made in hotels but are not being created on cultural sites. The virtual tour also corresponds with the need for personal hygiene. Such paid offers could be an answer for people who can not visit the sites not only because of pandemics but also due to individual conditions. Such offers could also be directed to distant schools as teaching materials. Another universal solution is creating a multicultural social portal where managers of cultural centers could exchange good practices and adapt their needs along with the current expectations of their clients. It would be an excellent exchange of information and demands.
Reviewer 2 Report
I cannot recommend this paper in this form. The structure is not clear. Important parts are missing (Introduction, Literature review etc.). Citations format is not regular. My decision is: Reconsider after major revision.
1. What is the main question addressed by the research? This study documents some customer behavior in the travel services market before and during the COVID-19 Pandemic (C-19). They offer theory-based and research-based insights that demonstrate customer value propositions during a pandemic and help predict future behavior for green tourism development. Comment: Literature review is needed. 2. Do you consider the topic original or relevant in the field? Does it address a specific gap in the field? Yes, topic is relevant. 3. What does it add to the subject area compared with other published material? This paper examines the cross-border cooperation from the above-mentioned aspects. 4. What specific improvements should the authors consider regarding the methodology? What further controls should be considered? More clear methodology must be presented. A simple data analysis is not enough. 5. Are the conclusions consistent with the evidence and arguments presented and do they address the main question posed? None. Crystall clear evidences and arguments mast be presented. 6. Are the references appropriate? None. Not enough, the format is not regular, not numbered. Basic citations are missing. E.g. https://cerphg.unideb.hu/PDF/2011_2/bujdoso.pdf and http://gtg.webhost.uoradea.ro/PDF/GTG-2-2015/3_178_Lorant.pdf and Dávid Lóránt, Bujdosó Zoltán, Tóth Géza Tourism planning in the Hajdú-Bihar – Bihor Euroregion In: Süli-Zakar, I (szerk.) Neighbours and partners : on the two sides of the border Debrecen, Magyarország : Kossuth Egyetemi Kiadó (2008) 402 p. pp. 323-332. , 10 p. https://www.worldcat.org/search?q=isbn%3A9789634731702 Etc. 7. Please include any additional comments on the tables and figures. Two figures are not enough. The second one is not numbered. Tables are relevant.
Author Response
Response to Reviewer Comments
In the beginning, the authors of the following article would like to thank you very much for your valuable comments and tips. Their inclusion undoubtedly raised the quality of the paper significantly. Below are the main changes made.
According to the tips, the authors revised the Introdution and changed the article's main section ( 1. Introduction, 2. Material and Methods, 3. Discussion, 4. Analysis of the market situation of both studied objects in the period preceding the pandemic and during the pandemic, 5. Value propositions for the client during the pandemic introduced by the managers at Książ Castle and the Kuks Complex, 6. Conclusions. Such a division seems to be more structured and logical for the reader.
Point 1: What is the main question addressed by the research?
Response 1: To better organize our paper, we changed the section Methodology to Material and Methodology. In this section, we underlined the six research problems and eight research questions, hoping it will provide a kind of roadmap for the reader.
(p. 8) The research problems focused on the following:
- Determining the quantitative relationship between pandemic-related insults at different times of the pandemic and the level of ticket sales;
- Identifying the response of businesses to changes resulting from the pandemic in general and concerning green economy solutions;
- Identifying the level of customer satisfaction with the changes made;
- Identifying the added value accepted and postulated by customers of both monuments;
- Identifying differences in the practice of the two monuments in two different countries during the pandemic;
- Identifying the added value for the customer in both monuments. As a recommendation for other such facilities in Poland and the Czech Republic concerning the green economy.
In connection with the development of the problem areas, the authors identified the following research questions:
- How did the level of visitors change with the pandemic at the two sites?
- How have changes in the level of ticket sales affected the overall operation of the two facilities?
- How was both facilities' adaptation and response process during the pandemic?
- What measures have been taken at both facilities to maintain financial stability in connection with the pandemic? What actions have been taken by both facilities to change competitive instruments (product, price, distribution channels, promotional activities)?
- How did both facilities react to their customers during the pandemic? With particular attention to solutions relating to the green economy, what resources did both facilities tap into?
- What lasting changes in buying behavior and expectations did the pandemic cause among customers of historic sites?
- What added value did both venues introduce during the pandemic, and what added value did their customers expect?
- What recommendations can be defined for similar facilities in both countries at the level of such crises emerging? And which recommendations can be adopted permanently as a continuous improvement of their services?
Point 2: This study documents some customer behavior in the travel services market before and during the COVID-19 Pandemic (C-19). They offer theory-based and research-based insights that demonstrate customer value propositions during a pandemic and help predict future behavior for green tourism development. Comment: Literature review is needed.
Response 2: In the Material and Methodology section, we enriched the literature by adding references to the theory. We used 29 new literature positions to make it more valuable for the readers. Below are examples of a few literature positions used in the paper.
- Stephen Espiner, Caroline Orchiston & James Higham (2017) Resilience and sustainability: a complementary relationship? Towards a practical conceptual model for the sustainability–resilience nexus in tourism, Journal of Sustainable Tourism, 25:10, 1385-1400, DOI:10.1080/09669582.2017.1281929
- Wassler Ph., Fan D.XF (2021), A tale of four futures: Tourism academia and COVID-19, Tourism Management Perspectives, Volume 38, April 2021, 100818, https://doi.org/10.1016/j.tmp.2021.100818, access: 24.01.2023.
- Ahmed, R. R., Streimikiene, D., Rolle, J-A, & Duc, P. A. (2020). The COVID-19 Pandemic and the
Antecedents for the Impulse Buying Behavior of US Citizens. Journal of Competitiveness, 12(3), 5–27.
https://doi.org/10.7441/joc.2020.03.01
- Zenker, Kock 2020a, b.. Zenker, S., Kock, F. 2020. The coronavirus pandemic - A critical discussion of a tourismresearch agenda. Tourism Management, 81, Article 104164, https://www.ncbi.nlm.nih.gov/pmc/articles/PMC7272331/ access: 20.01.2023
- Godovykh M., Ridderstaat J., Health outcomes of tourism development: A longitudinal study of the impact of tourism arrivals on residents’ health. Journal of Destination Marketing & Management, Volume 17, September 2020, 100462, https://www.sciencedirect.com/science/article/pii/S2212571X20300846?casa_token=kB8bS7ycfrQAAAAA:yJ4IFyZyjRKI90_E_XI6qAr3GkbkZdUke63-kl66moRocuUkCe95EcRA-K8PJWnErA8QysQAzPY, access: 24.01.2023.
- Abu Bakar, N., & Rosbi, S., (2020). Effect of Coronavirus disease (COVID-19) to the tourism industry. International Journal of Advanced Engineering Research and Science, 7(4), pp. 189-193, https://dx.doi.org/10.22161/ijaers.74.23.
Point 3 and 4: Do you consider the topic original or relevant in the field? Does it address a specific gap in the field? Yes, topic is relevant.
Response 3 and 4: Introduction. We refilled it with information about why the topic is essential and what is the research gap. We also presented the added values, which might be the future way for the touristic branch.
"The topic is important because the situation during the pandemic showed the lack of a quick response, which is only possible if you have prepared scenarios for the crisis. Modern times are often defined as a time of permanent crisis. Therefore, it is crucial for those running companies today, including heritage managers, to constantly observe how to improve and create new added value to retain and/or acquire customers. On the other hand, the research gap is to identify those added values for clients of historical sites that should be introduced regardless of the pandemic period due to permanent changes in client behavior and expectations. The added values that the authors observed, as activities introduced or desired at both historic sites not only during the pandemic, are:
- Increasing the safety of personal hygiene and interpersonal contacts through the introduction of tools such as the development of a particular instruction manual;
- Paying attention to green management, which resulted from savings, as well as customer expectations.
- Using tools from the area of digitization both when selling services and enriching the product, distribution, and promotion of services in the two historic sites.
In the coronavirus era, the consumer's attitude has changed because his activities related to market activity were influenced by a different way of thinking and assessing the situation. The current lifestyle, focusing on the problems of frequency of shopping, participation in social circles, entertainment, recreation, hobbies, or the very attitude to work, had to change, regardless of the will of a given person. This surprising situation changed customers' personalities, which, as it was found, under stable conditions, changes over the years, and today's time has accelerated and changed us, the environment, and even our life goals, plans, and priorities. Marketing changes behaviors, habits, views of individuals and social groups, and the value system under the influence of specific activities, called marketing activities. Practical marketing activities are now considered to have a real, measurable impact on the financial result, profitability, or value of the enterprise, especially by increasing the total customer value (Woźniczka et al., 2014, p. 18). New marketing trends are reflected in social, cultural, and ecological aspects resulting from buyer behavior changes, which new organizations should remember. As shown by market laws, enterprises cannot only focus on business goals but must also consider the principles of the green economy.
Point 5: What specific improvements should the authors consider regarding the methodology? What further controls should be considered? More clear methodology must be presented. A simple data analysis is not enough.
Response 5: To clarify the authors' methods in their study, we added a sentence: (p.8). The main objective of the conducted research was to identify the added value that was generated or should be implemented by companies managing historical sites in both Poland and the Czech Republic - during the pandemic based on: internal documents of both organizations, analyses of the level of tourist traffic, interviews with managers and surveys among tourists, which both companies carry out on an ongoing basis. The authors also conducted their research through face-to-face interviews among tourists at both facilities. The questionnaire took 132 respondents between June and August 2020 at both facilities
During the study, the authors used the following research methods: simple statistical methods, comparative analysis, analysis of foundational documents in both entities, interviews with the management of both facilities, and discussion and analysis of surveys conducted
Point 6: Are the conclusions consistent with the evidence and arguments presented and do they address the main question posed? None. Crystall clear evidences and arguments mast be presented.
Response 6: To make the conclusions more consistent with the evidence and arguments presented, we decided to add the discussion section to the paper.
(pp.9-10) Most of the available research results on consumer behavior resulting from the onset of a pandemic concern the first few months after the outbreak. They cover selected aspects of purchasing behavior in various product categories. The results of studies conducted in the early period of the pandemic indicated that consumers changed their behavior and sharply increased their purchases of certain products (Aydınlıoğlu, Gencer, 2020; Islam et al., 2020; Laato et al., 2020). As M. Loxton et al. (2020) point out, consumers in the first months of the epidemic behaved similarly to previous crises, such as during the 2007-2009 financial crisis - they bought products in a panic, reduced spending on elective goods and made their decisions based on media reports
The COVID-19 pandemic has significantly affected tourism. Many places that have enjoyed enduring popularity for years saw a significant drop in visitors in 2020. The introduction of restrictions in the form of a ban on movement has contributed to a substantial decline in tourist arrivals in Poland and the Czech Republic. The literature describes the changes that COVID-19 has caused in the tourism economy (Goodgrer and Kieran, 2020), formulates multivariate forecasts of the development of the situation, draws scenarios for recovery from the crisis, and proposes corrective measures. A special issue of Tourism Geographies (vol. 22, no. 3) was published, containing as many as 25 articles on COVID-19-related topics, for example, outlining a vision for post-emergency tourism (Haywood, 2020), proposing necessary economic measures to save the tourism economy (Cave and Dredge, 2020), and pointing to the required transformation (Hall, Scott, and Gössling, 2020). There are theses about the need to socialize and green tourism after COVID-19 (Higgins-Desbiolles, 2020; Crossley, 2020) with a strong emphasis on sustainability and responsible tourism (Niewiadomski, 2020). It has been proven that during recovery from an epidemic crisis, traveling within one's own country and staying at agritourism farms and facilities to ensure sanitary safety will become more popular (Borodako, 2021). In the era of pandemics, business models in tourism, in the area of sights, referred to the opportunities offered by the virtual world. We are seeing more and more discussions related to changes in the progressive digitization of sales processes (Paschen et al., 2020b; Syam and Sharma, 2017). Researchers recognize a gap in the area of understanding the impact of Artificial Intelligence (AI), including Machine Learning (ML) and Digital Marketing (DM), on modern sales (Lilien, 2016; Paschen et al., 2020a). The literature contains numerous studies on the distinction between marketing and sales (Panagopoulos and Avlonitis, 2010).
In contrast, research on how consumer behavior changes affect the sales function's remodeling relative to the marketing function is an episodic phenomenon (Johnson, 2015). Currently, the biggest challenge facing businesses, management trends 62 e-mentor No. 4 (91), is the COVID-19 pandemic, which affects all business areas, including the sales function (Bothe et al., 2021). A pandemic outbreak is part of a broader problem of various phenomena referred to as exogenous shocks, which tend to have a jumping effect on changes in economic and social processes (Murawska, 2020). They are part of the peculiarities of standard economic shocks - demand, supply, price, and financial shocks, which have affected almost all industries, forcing companies to reformulate their implemented strategies, including sales strategies (Geiger and Guenzi, 2009; Panagopoulos and Avlonitis, 2010; Radlinska, 2020; Rangarajan et al., 2018). Knowledge in this area is drawn mainly from commercial publications, consulting firms' reports, and practitioners' studies (Lane and Piercy, 2009; Leigh and Marshall, 2001; Palmer and Flanagan, 2016). Overall, during the COVID-19 pandemic, a theory can be advanced that the tourism system is more vulnerable than other systems [Espiner et al. 2017]. The pandemic caused unprecedented effects: it affected tourists' lifestyles, behaviors, and travel patterns [Wassler, Fan 2021]. In addition, an overall increase in mental disorders caused by isolation, such as increased anxiety, impacted the frequency and form of tourism participation [Ahmed et al. 2020; Kock et al. 2020]. The pandemic also resulted in other adverse consequences, such as creating a negative image of travel [Godovykh, Ridderstaat 2020]. Quite a bit of change could also be seen in the destination's vision, which, as is well known, can change constantly. The COVID-19 pandemic significantly impacted its further formation [Zenker, Kock 2020]. Overall, it affected global supply and demand [Abu Bakar, Rosbi, 2020]. Hence, it is understandable that there has been a marked increase in interest in the issue of tourism crisis and crisis management [Sigala 2020; Baum, Hai 2020; Hall et al. 2020], among others, in the context of pandemic mitigation and post-crisis recovery of the industry [Yeh 2021].
In the Polish literature, one can find studies of a pilot and non-representative nature (Gorzelany--Dziadkowiec, 2020; Samuk, Sidorowicz, 2020). However, they allow us to see symptoms of changes in the structure of spending (reduction of clothing and footwear purchases), increased online shopping activity, and the use of instant messaging in the purchasing process. A change in consumer attitudes has been identified: the growing importance of rest, relaxation, and a healthy lifestyle. In the pages of the journal Organization Review, the results of the study of consumer behavior in light of COVID-19 have not yet been published, but the issues of sustainability of behavioral changes under the influence of modification of marketing activities of enterprises and habits of buyers have been considered (Brzezinski, 2020). The author of the cited article put forward the thesis that the changes observed during the pandemic will be permanent.
When considering the tourist attractions of the Polish-Czech border region as a green economy initiative, it is necessary to point out what the green economy is. The concept has emerged in debates about sustainable development, especially in light of the Rio+20 idea [United Nations Conference on Environment and Development meeting in Rio de Janeiro from June 3 to 14, 1992]. For the aims of this article, it has been assumed that the green economy generally refers to an economy in which the quality of human life and the state of the environment are considered broadly when making production and consumption decisions. Hence, the inclusion of tourist attractions of the Polish-Czech borderland in this concept/initiative should be considered a novelty.posiedzeniu w Rio de Janeiro w dniach od 3 do 14 czerwca 1992 r.].
Conducted in both the literature and own research in the form of face-to-face interviews with the managers of the analyzed monuments, as well as obtained answers in a survey conducted among visitors to the monuments, the authors formulated answers to the research questions posed in the Material and Methods section. The added value that the authors observed, as actions introduced or desirable in both historic sites not only during the pandemic, are: 1. Increasing safety in the area of personal hygiene and human contact through the introduction of tools such as the development of special handling instructions; 2. They are paying attention to green management, which results from savings and customer expectations. 3. using tools from the area of digitization both when selling.
Point 7: Are the references appropriate? None. Not enough, the format is not regular, not numbered. Basic citations are missing. E.g. https://cerphg.unideb.hu/PDF/2011_2/bujdoso.pdf and http://gtg.webhost.uoradea.ro/PDF/GTG-2-2015/3_178_Lorant.pdf and Dávid Lóránt, Bujdosó Zoltán, Tóth Géza Tourism planning in the Hajdú-Bihar – Bihor Euroregion In: Süli-Zakar, I (szerk.) Neighbours and partners : on the two sides of the border Debrecen, Magyarország : Kossuth Egyetemi Kiadó (2008) 402 p. pp. 323-332. , 10 p. https://www.worldcat.org/search?q=isbn%3A9789634731702 Etc.
Response 8-7: We increased our work by 29 new items and formatted the bibliography according to the guidelines. We verified all the websites we provided.
Point 8: Please include any additional comments on the tables and figures. Two figures are not enough. The second one is not numbered. Tables are relevant
Response 8: The quality of graphical/illustrative representations has been improved.
Reviewer 3 Report
The research topic is very relevant, but the structure of the article is flawed and not typical for a scientific article. Authors must note the following comments.
Abstract:
Authors must first outline the research problem, methodology (theoretical and empirical part, research methods and techniques), the added value of the research, and added value of the article from the professional/scientific point of view.
Chapter 1:
The authors must start by clearly stating the research problem and explaining why the research problem is worth exploring. In this chapter, it should be made clear what the added value of the research is and what the added value of the article is. The authors must present which of the research paradigms are going to be used (qualitative paradigm, quantitative paradigm, mixed). The authors must also present which of the mentioned research methods belongs to an individual paradigm and which of these research methods are going to be used in the theoretical and in the empirical part of his research. Authors must also outline stages/steps which were used in the empirical part from data collection to data analysis. At the same time, it must also be presented whether the authors faced any assumptions and limitations in collecting and analysing the data during research.
Conclusion:
Authors must answer the following question: What is the original contribution of the article?
The quality of graphical/illustrative representations must be improved, especially figure 1 which contains lapses.
Author Response
In the beginning, the authors of the following article would like to thank you very much for your valuable comments and tips. Their inclusion undoubtedly raised the quality of the paper significantly. Below are the main changes made.
Point 1: Abstract: Authors must first outline the research problem, methodology (theoretical and empirical part, research methods and techniques), the added value of the research, and added value of the article from the professional/scientific point of view.
According to the tips, the authors revised the Introdution and changed the article's main section ( 1. Introduction, 2. Material and Methods, 3. Discussion, 4. Analysis of the market situation of both studied objects in the period preceding the pandemic and during the pandemic, 5. Value propositions for the client during the pandemic introduced by the managers at Książ Castle and the Kuks Complex, 6. Conclusions. Such a division seems to be more structured and logical for the reader.
Point 1: research problem
Response 1: We changed the section Methodology to Material and Methodology to better organize our paper. In this section, we underlined the six research problems and eight research questions, hoping it will provide a kind of roadmap for the reader.
(p. 8) The research problems focused on the following:
- Determining the quantitative relationship between pandemic-related insults at different times of the pandemic and the level of ticket sales;
- Identifying the response of businesses to changes resulting from the pandemic in general and concerning green economy solutions;
- Identifying the level of customer satisfaction with the changes made;
- Identifying the added value accepted and postulated by customers of both monuments;
- Identifying differences in the practice of the two monuments in two different countries during the pandemic;
- Identifying the added value for the customer in both monuments. As a recommendation for other such facilities in Poland and the Czech Republic concerning the green economy.
In connection with the development of the problem areas, the authors identified the following research questions:
- How did the level of visitors change with the pandemic at the two sites?
- How have changes in the level of ticket sales affected the overall operation of the two facilities?
- How was both facilities' adaptation and response process during the pandemic?
- What measures have been taken at both facilities to maintain financial stability in connection with the pandemic? What actions have been taken by both facilities to change competitive instruments (product, price, distribution channels, promotional activities)?
- How did both facilities react to their customers during the pandemic? With particular attention to solutions relating to the green economy, what resources did both facilities tap into?
- What lasting changes in buying behavior and expectations did the pandemic cause among customers of historic sites?
- What added value did both venues introduce during the pandemic, and what added value did their customers expect?
- What recommendations can be defined for similar facilities in both countries at the level of such crises emerging? And which recommendations can be adopted permanently as a continuous improvement of their services?
Point 2 methodology (theoretical and empirical part, research methods and techniques)
Response 2: To clarify the authors' methods in their study, we added a sentence: (p.8). The main objective of the conducted research was to identify the added value that was generated or should be implemented by companies managing historical sites in both Poland and the Czech Republic - during the pandemic based on: internal documents of both organizations, analyses of the level of tourist traffic, interviews with managers and surveys among tourists, which both companies carry out on an ongoing basis. The authors also conducted their research through face-to-face interviews among tourists at both facilities. The questionnaire took 132 respondents between June and August 2020 at both facilities
During the study, the authors used the following research methods: simple statistical methods, comparative analysis, analysis of foundational documents in both entities, interviews with the management of both facilities, and discussion and analysis of surveys conducted
We added the discussion section to the paper to make the conclusions more consistent with the evidence and arguments presented.
(pp.9-10) Most of the available research results on consumer behavior resulting from the onset of a pandemic concern the first few months after the outbreak. They cover selected aspects of purchasing behavior in various product categories. The results of studies conducted in the early period of the pandemic indicated that consumers changed their behavior and sharply increased their purchases of certain products (Aydınlıoğlu, Gencer, 2020; Islam et al., 2020; Laato et al., 2020). As M. Loxton et al. (2020) point out, consumers in the first months of the epidemic behaved similarly to previous crises, such as during the 2007-2009 financial crisis - they bought products in a panic, reduced spending on elective goods and made their decisions based on media reports
The COVID-19 pandemic has significantly affected tourism. Many places that have enjoyed enduring popularity for years saw a significant drop in visitors in 2020. The introduction of restrictions in the form of a ban on movement has contributed to a substantial decline in tourist arrivals in Poland and the Czech Republic. The literature describes the changes that COVID-19 has caused in the tourism economy (Goodgrer and Kieran, 2020), formulates multivariate forecasts of the development of the situation, draws scenarios for recovery from the crisis, and proposes corrective measures. A special issue of Tourism Geographies (vol. 22, no. 3) was published, containing as many as 25 articles on COVID-19-related topics, for example, outlining a vision for post-emergency tourism (Haywood, 2020), proposing necessary economic measures to save the tourism economy (Cave and Dredge, 2020), and pointing to the required transformation (Hall, Scott, and Gössling, 2020). There are theses about the need to socialize and green tourism after COVID-19 (Higgins-Desbiolles, 2020; Crossley, 2020) with a strong emphasis on sustainability and responsible tourism (Niewiadomski, 2020). It has been proven that during recovery from an epidemic crisis, traveling within one's own country and staying at agritourism farms and facilities to ensure sanitary safety will become more popular (Borodako, 2021). In the era of pandemics, business models in tourism, in the area of sights, referred to the opportunities offered by the virtual world. We are seeing more and more discussions related to changes in the progressive digitization of sales processes (Paschen et al., 2020b; Syam and Sharma, 2017). Researchers recognize a gap in the area of understanding the impact of Artificial Intelligence (AI), including Machine Learning (ML) and Digital Marketing (DM), on modern sales (Lilien, 2016; Paschen et al., 2020a). The literature contains numerous studies on the distinction between marketing and sales (Panagopoulos and Avlonitis, 2010).
In contrast, research on how consumer behavior changes affect the sales function's remodeling relative to the marketing function is an episodic phenomenon (Johnson, 2015). Currently, the biggest challenge facing businesses, management trends 62 e-mentor No. 4 (91), is the COVID-19 pandemic, which affects all business areas, including the sales function (Bothe et al., 2021). A pandemic outbreak is part of a broader problem of various phenomena referred to as exogenous shocks, which tend to have a jumping effect on changes in economic and social processes (Murawska, 2020). They are part of the peculiarities of standard economic shocks - demand, supply, price, and financial shocks, which have affected almost all industries, forcing companies to reformulate their implemented strategies, including sales strategies (Geiger and Guenzi, 2009; Panagopoulos and Avlonitis, 2010; Radlinska, 2020; Rangarajan et al., 2018). Knowledge in this area is drawn mainly from commercial publications, consulting firms' reports, and practitioners' studies (Lane and Piercy, 2009; Leigh and Marshall, 2001; Palmer and Flanagan, 2016). Overall, during the COVID-19 pandemic, a theory can be advanced that the tourism system is more vulnerable than other systems [Espiner et al. 2017]. The pandemic caused unprecedented effects: it affected tourists' lifestyles, behaviors, and travel patterns [Wassler, Fan 2021]. In addition, an overall increase in mental disorders caused by isolation, such as increased anxiety, impacted the frequency and form of tourism participation [Ahmed et al. 2020; Kock et al. 2020]. The pandemic also resulted in other adverse consequences, such as creating a negative image of travel [Godovykh, Ridderstaat 2020]. Quite a bit of change could also be seen in the destination's vision, which, as is well known, can change constantly. The COVID-19 pandemic significantly impacted its further formation [Zenker, Kock 2020]. Overall, it affected global supply and demand [Abu Bakar, Rosbi, 2020]. Hence, it is understandable that there has been a marked increase in interest in the issue of tourism crisis and crisis management [Sigala 2020; Baum, Hai 2020; Hall et al. 2020], among others, in the context of pandemic mitigation and post-crisis recovery of the industry [Yeh 2021].
In the Polish literature, one can find studies of a pilot and non-representative nature (Gorzelany--Dziadkowiec, 2020; Samuk, Sidorowicz, 2020). However, they allow us to see symptoms of changes in the structure of spending (reduction of clothing and footwear purchases), increased online shopping activity, and the use of instant messaging in the purchasing process. A change in consumer attitudes has been identified: the growing importance of rest, relaxation, and a healthy lifestyle. In the pages of the journal Organization Review, the results of the study of consumer behavior in light of COVID-19 have not yet been published, but the issues of sustainability of behavioral changes under the influence of modification of marketing activities of enterprises and habits of buyers have been considered (Brzezinski, 2020). The author of the cited article put forward the thesis that the changes observed during the pandemic will be permanent.
When considering the tourist attractions of the Polish-Czech border region as a green economy initiative, it is necessary to point out what the green economy is. The concept has emerged in debates about sustainable development, especially in light of the Rio+20 idea [United Nations Conference on Environment and Development meeting in Rio de Janeiro from June 3 to 14, 1992]. For the aims of this article, it has been assumed that the green economy generally refers to an economy in which the quality of human life and the state of the environment are considered broadly when making production and consumption decisions. Hence, the inclusion of tourist attractions of the Polish-Czech borderland in this concept/initiative should be considered a novelty.posiedzeniu w Rio de Janeiro w dniach od 3 do 14 czerwca 1992 r.].
Points 3 and 4: the added value of the research, and added value of the article from the professional/scientific point of view
Response 3 and 4: Introduction. We refilled it with information about why the topic is essential and what is the research gap. We also presented the added values, which might be the future way for the touristic branch.
"The topic is important because the situation during the pandemic showed the lack of a quick response, which is only possible if you have prepared scenarios for the crisis. Modern times are often defined as a time of permanent crisis. Therefore, it is crucial for those running companies today, including heritage managers, to constantly observe how to improve and create new added value to retain and/or acquire customers. On the other hand, the research gap is to identify those added values for clients of historical sites that should be introduced regardless of the pandemic period due to permanent changes in client behavior and expectations. The added values that the authors observed, as activities introduced or desired at both historic sites not only during the pandemic, are:
- Increasing the safety of personal hygiene and interpersonal contacts through the introduction of tools such as the development of a particular instruction manual;
- Paying attention to green management, which resulted from savings, as well as customer expectations.
- Using tools from the area of digitization both when selling services and enriching the product, distribution, and promotion of services in the two historic sites.
In the coronavirus era, the consumer's attitude has changed because his activities related to market activity were influenced by a different way of thinking and assessing the situation. The current lifestyle, focusing on the problems of frequency of shopping, participation in social circles, entertainment, recreation, hobbies, or the very attitude to work, had to change, regardless of the will of a given person. This surprising situation changed customers' personalities, which, as it was found, under stable conditions, changes over the years, and today's time has accelerated and changed us, the environment, and even our life goals, plans, and priorities. Marketing changes behaviors, habits, views of individuals and social groups, and the value system under the influence of specific activities, called marketing activities. Practical marketing activities are now considered to have a real, measurable impact on the financial result, profitability, or value of the enterprise, especially by increasing the total customer value (Woźniczka et al., 2014, p. 18). New marketing trends are reflected in social, cultural, and ecological aspects resulting from buyer behavior changes, which new organizations should remember. As shown by market laws, enterprises cannot only focus on business goals but must also consider the principles of the green economy.
Moreover, the researchers recognize a gap in the area of understanding the impact of Artificial Intelligence (AI), including Machine Learning (ML) and Digital Marketing (DM), on modern sales (Lilien, 2016; Paschen et al., 2020a). The literature contains numerous studies on the distinction between marketing and sales (Panagopoulos and Avlonitis, 2010)
Point 5 Conclusion:
Authors must answer the following question: What is the original contribution of the article?
Response 5: The added value that the authors observed, as actions introduced or desirable in both historic sites not only during the pandemic, are: 1. Increasing safety in the area of personal hygiene and human contact through the introduction of tools such as the development of special handling instructions; 2. They are paying attention to green management, which results from savings and customer expectations. 3. Using tools from the area of digitization both when selling services as well as enriching the product, distribution, and promotion of services in the two historic sites.
According to obtained added value, the authors consider that one of the study's main conclusions is using and developing artificial intelligence (AI), which would enable a virtual tour of the monuments. Such offers have been made in hotels but are not being created on cultural sites. The virtual tour also corresponds with the need for personal hygiene. Such paid offers could be an answer for people who can not visit the sites not only because of pandemics but also due to individual conditions. Such offers could also be directed to distant schools as teaching materials. Another universal solution is creating a multicultural social portal where managers of cultural centers could exchange good practices and adapt their needs along with the current expectations of their clients. It would be an excellent exchange of information and demands.
Point 6: The quality of graphical/illustrative representations must be improved, especially figure 1 which contains lapses.
Response 6: The quality of graphical/illustrative representations has been improved
Round 2
Reviewer 1 Report
The Authors revised the paper in line with al suggestions and comments. I find the paper ready for publishing.
Author Response
Good morning,
Thank you again for your valuable comments.
kind regards
Małgorzata Pol
Reviewer 2 Report
This citation is not correct:
Bujdosó Zoltán , Tóth Géza Planowanie turystyki w euroregionie Hajdú-Bihar – Bihor W: Süli- 903 Zakar, I (szerk.) SÄ…siedzi i partnerzy: po obu stronach granicy Debrecen, 904 Magyarország: Kossuth Egyetemi Kiadó (2008) 402 pp.. 323-332
Correct:
Dávid Lóránt, Bujdosó Zoltán, Tóth Géza
Tourism planning in the Hajdú-Bihar – Bihor Euroregion In: Süli-Zakar, I (szerk.) Neighbours and partners : on the two sides of the border Debrecen, Magyarország : Kossuth Egyetemi Kiadó (2008) 402 p. pp. 323-332. , 10 p.
Please add this basic work to References:
Dávid Lóránt, Szűcs Csaba
Building of networking, clusters and regions for tourism in the Carpathian Basin via information and communication technologies NETCOM - NETWORKS AND COMMUNICATIONS STUDIES 23 : 1-2 pp. 63-74. , 12 p. (2009)
Author Response
Good morning
thank you for your comments.
All the mentioned comments have been taken into account
kind regards
Małgorzata Pol
Reviewer 3 Report
In conclusion, authors are required to answer all 8 research questions, which are presented in the Materials and Methods chapter.
Author Response
Thank you very much for your comments.
The conclusions included seemed sufficient; however, as suggested, we have supplemented our findings with an additional table that answers the research questions.
With best regards
Author
|
research questions |
Książ |
Kuks |
|
How did the level of visitors change with the pandemic at the two sites?
|
decreased by 23% |
decreased by over 40% |
|
How have changes in the level of ticket sales affected the overall operation of the two facilities?
|
E-marketing and e-sales activities have increased. |
Reduced activity in promoting the facility. |
|
How was both facilities' adaptation and response process during the pandemic?
|
Solutions to increase the safety of tourists were introduced very quickly |
After a few months, adjustment measures were introduced to improve the safety of tourists. |
|
What measures have been taken at both facilities to maintain financial stability in connection with the pandemic? |
Reduced revenues have halted investment and renovation activities |
The number of staff has been reduced, and current costs have been significantly reduced. |
|
What actions have been taken by both facilities to change competitive instruments (product, price, distribution channels, promotional activities)?
|
New products have been introduced that are linked to changes in price and distribution, such as a personalized tour with audio-guides |
The possibility of virtual tours in place of traditional tours has been developed. |
|
How did both facilities react to their customers during the pandemic? With particular attention to solutions relating to the green economy, what resources did both facilities tap into?
|
Revitalization of outdoor gardens and thermal upgrades inside the building. Thanks to this, the facility can apply to be a health resort. |
Thermal modernization of the building's heating system. In the future, reduce heating costs. |
|
What lasting changes in buying behavior and expectations did the pandemic cause among customers of historic sites?
|
Increase the share of individual tours without a guide. |
Reduction in the number of tourists visiting directly by eliminating visits to the main parts of the castle, expanding the number of virtual tourists. |
|
What added value did both venues introduce during the pandemic, and what added value did their customers expect?
|
introduced: individual tours expected: change in ticket prices |
introduced: virtual tours expected: opportunities for safe tours |
|
What recommendations can be defined for similar facilities in both countries at the level of such crises emerging? And which recommendations can be adopted permanently as a continuous improvement of their services?
|
After a decline in the months of III - V, there was a sharp increase in June and July 2020, then stability was observed. |
The decline was pronounced until autumn 2020; restoration of visitor counts occurred from spring 2021. |